# Secondary Organic Aerosol Formation from Camphene Oxidation: Measurements and Modeling

Qi Li[1,2], Jia Jiang[1,2], Isaac Kwadjo Afreh [1,2], Kelley C. Barsanti[1,2], David R. Cocker III[1,2]

[1]Department of Chemical and Environmental Engineering, University of California-Riverside, Riverside, California 92521, United States
[2]The Bourns College of Engineering, Center for Environmental Research and Technology, University of California-Riverside, Riverside, California 92507, United States

*Correspondence to*: Kelley C. Barsanti (kbarsanti@engr.ucr.edu) and David R. Cocker III (dcocker@engr.ucr.edu)

**Abstract.** While camphene is one of the dominant monoterpenes measured in biogenic and pyrogenic emissions samples, oxidation of camphene has not been well-studied in environmental chambers and very little is known about its potential to form secondary organic aerosol (SOA). The lack of chamber-derived SOA data for camphene may lead to significant uncertainties in predictions of SOA from oxidation of monoterpenes using existing parameterizations when camphene is a significant contributor to total monoterpenes. Therefore, to advance the understanding of camphene oxidation and SOA formation, and to improve representation of camphene in air quality models, a series of experiments were performed in the University of California Riverside environmental chamber to explore camphene SOA mass yields and properties across a range of chemical conditions at atmospherically relevant OH concentrations. The experimental results were compared with modeling simulations obtained using two chemically detailed box models, Statewide Air Pollution Research Center (SAPRC) and Generator for Explicit Chemistry and Kinetics of Organics in the Atmosphere (GECKO-A). SOA parameterizations were derived from the chamber data using both the two-product and volatility basis set (VBS) approaches. Experiments performed with added nitrogen oxides ($NO_x$) resulted in higher SOA mass yields (up to 64%) than experiments performed without added $NO_x$ (up to 28%). In addition, camphene SOA mass yields increased with SOA mass ($M_o$) at lower mass loadings, but a threshold was reached at higher mass loadings in which the SOA mass yields no longer increased with $M_o$. SAPRC modeling of the chamber studies suggested that the higher SOA mass yields at higher initial $NO_x$ levels were primarily due to higher production of peroxy radicals ($RO_2$) and the generation of highly oxygenated organic molecules (HOMs) formed through unimolecular $RO_2$ reactions. SAPRC predicted that in the presence of $NO_x$, camphene $RO_2$ reacts with NO and the resultant $RO_2$ undergo hydrogen (H)-shift isomerization reactions; as has been documented previously, such reactions rapidly add oxygen and lead to products with very low volatility (i.e., HOMs). The end products formed in the presence of $NO_x$ have significantly lower volatilities, and higher O:C ratios, than those formed by initial camphene $RO_2$ reacting with hydroperoxyl radicals ($HO_2$) or other $RO_2$. Further analysis reveals the existence of an extreme $NO_x$ regime, where the SOA mass yield can be suppressed again due to high $NO/HO_2$ ratios. Moreover, particle densities were found to decrease from 1.47 to 1.30 g cm$^{-3}$ as $[HC]_0/[NO_x]_0$ increased and O:C decreased. The observed differences in SOA mass yields were largely explained by the gas-phase $RO_2$ chemistry and the competition between $RO_2 + HO_2$, $RO_2 + NO$, $RO_2 + RO_2$, and $RO_2$ autoxidation reactions.

## 1 Introduction

On a global scale, biogenic monoterpene emissions are estimated to contribute 14% of the total reactive volatile organic compound (VOC) flux (Tg C) (Guenther, 1995). Camphene is an ubiquitous monoterpene emitted from biogenic sources (Geron et al., 2000; Hayward et al., 2001; Ludley et al., 2009; Maleknia et al., 2007; White et al., 2008) and pyrogenic sources ( Akagi et al., 2013; Gilman et al., 2015; Hatch et al., 2015). Many studies have reported camphene as a top contributor by mass in measured biogenic and pyrogenic monoterpene emissions (Benelli et al., 2018; Hatch et al., 2019; Komenda, 2002; Mazza & Cottrell, 1999; Moukhtar et al., 2006). For example, in measurements of laboratory and prescribed fires reported by Hatch et al. (2019), camphene was among the top two monoterpenes emitted from subalpine and Douglas fir fires based on emission factors (mass of compound emitted/mass of fuel burned).

**Figure 1. Camphene chemical structure and reaction rate constants (unit: $cm^3$ $molecule^{-1}$ $s^{-1}$) with major atmospheric oxidants.**

When emitted to the atmosphere, monoterpenes form oxygenated compounds through reactions with oxidants such as hydroxyl radicals (OH), ozone ($O_3$) and nitrate radicals ($NO_3$); compounds with sufficiently low volatility can then condense to form secondary organic aerosol (SOA). Figure 1 shows the chemical structure of camphene and its reaction rate constants with major atmospheric oxidants. The SOA formation potential of individual monoterpenes can vary greatly based on their molecular structure, atmospheric lifetimes, and the volatility of their oxidation products (Atkinson and Arey, 2003; Griffin et al., 1999; Ng et al., 2007a; Zhang et al., 1992). Previous experimental studies of other monoterpenes (such as α-pinene, β-pinene, d-limonene, etc.) have reported SOA mass yields from ~10% to 50% through OH oxidation and from ~ 0 to 65% through $NO_3$ oxidation; among the studied monoterpenes, d-limonene often has the highest reported yields (Mutzel et al., 2016; Griffin et al., 1999; Ng et al., 2007b; Fry et al., 2014). Few studies have been published regarding camphene SOA formation.

Past experimental studies of camphene largely have been focused on gas-phase reactivity with OH, $NO_3$, and/or $O_3$ and gas-phase product identification (e.g., Atkinson et al., 1990; Gaona-Colmán et al., 2017; Hakola et al., 1994). Baruah et al. (2018) performed a kinetic and mechanism study of the camphene oxidation initiated by OH radicals using density functional theory (DFT), in which the rate constant and atmospheric lifetime were reported. It was also suggested that addition at the terminal double bond carbon atom could account for 98.4% of the initial OH-addition. A product study by Gaona-Colmán et al. (2017) showed obvious $NO_x$ dependence in OH + camphene experiments, in

which the molar yield of acetone was enhanced by a factor of 3, 33% relative to 10%, in the presence of $NO_x$ (2–2.3 ppmv of NO).

Hatfield and Huff-Hartz studied SOA formation from ozonolysis of VOC mixtures, in which the added camphene was considered a non-reactive VOC and assumed to have little to no effect on SOA mass yields (Hatfield & Hartz, 2011). Mehra et al. (2020) recently published a compositional analysis study of camphene SOA. Although SOA mass yields were not provided, they demonstrated the potential contribution of highly oxygenated organic molecules (HOMs) and oligomers to camphene SOA formed in an oxidation flow reactor (OFR). Afreh et al. (2020) presented the first mechanistic modeling study of camphene SOA formation. While relatively high SOA mass yields were reported (with final SOA mass and yields twice that of α-pinene), no chamber-based SOA data were available for measurement–model comparison at that time.

SOA formation has been shown to be highly dependent on gas-phase $NO_x$ concentrations; and more precisely, the relative ratios of NO:$HO_2$, hydroperoxyl radicals:$RO_2$, peroxy radicals (Henze et al., 2008; Ng et al., 2007b; Presto et al., 2005; Ziemann and Atkinson, 2012; Kroll and Seinfeld, 2008; Song et al., 2005). During chamber experiments, VOCs are subject to oxidation by OH, $O_3$ and/or $NO_3$. For some precursors, $NO_x$ levels influence the amount of SOA produced in the initial oxidation steps by controlling the relative proportions of oxidants, the fractional reactivity with those oxidants, and thus the volatility distribution of the products formed (Hurley et al., 2001; Nøjgaard et al., 2006; Kroll and Seinfeld, 2008). For other precursors, $NO_x$ levels influence the amount of SOA produced via fate of $RO_2$. The reactions between $RO_2$ and $HO_2$ form hydroperoxides, which can have sufficiently low volatility to condense into the particle phase. In the presence of $NO_x$, $RO_2$ will react with NO, forming organic nitrate and carbonyl compounds that have higher volatilities than the products formed through the $HO_2$ pathway (Kroll and Seinfeld, 2008; Ziemann and Atkinson, 2012). Previous studies of relatively small compounds (carbon number ≤10), including monoterpenes such as α-pinene, have reported that SOA mass yields generally increase as initial $NO_x$ decreases, with a proposed mechanism of competitive chemistry between $RO_2 + HO_2$ and $RO_2 + NO$ pathways, of which the latter would form more volatile products (Kroll et al., 2006; Ng et al., 2007; Song et al., 2005). The $NO_x$ dependence of camphene oxidation and SOA formation has been relatively understudied.

The atmospheric gas-phase autoxidation of $RO_2$ has been identified as another key pathway of SOA formation (Crounse et al., 2013; Jokinen 2014; Ehn et al., 2017; Bianchi et al., 2019). The $RO_2$ radical undergoes intramolecular H-atom abstraction reactions to form a hydroperoxide functionality and an alkyl radical (RO), to which a new $RO_2$ will be formed by adding $O_2$. The autoxidation process can repeat several times until terminated by other pathways and will form low-volatility compounds known as highly oxygenated organic molecules (HOMs) (Bianchi et al., 2019). Recent theoretical and experimental studies have been conducted to understand HOM formation from monoterpenes such as α-pinene and β-pinene (Zhang et al., 2017; Quéléver et al., 2019; Xavier et al., 2019; Pullinen et al., 2020; Ye et al., 2020), but the potential importance and mechanisms of HOM formation from camphene have not been well investigated.

Here, we present the first systematic study of SOA formation from camphene using laboratory-based chamber experiments and chemically detailed box models. The experiments were conducted at varying $NO_x$ levels and the chamber data were used to provide SOA parameterizations based on the two-product (Odum et al., 1996) and volatility

basis set (VBS) modeling approaches (Donahue et al., 2006; Donahue et al., 2009). Two chemically detailed box models, Statewide Air Pollution Research Center (SAPRC) and Generator for Explicit Chemistry and Kinetics of Organics in the Atmosphere (GECKO-A), were used to provide mechanistic insights into the chamber observations and to elucidate the connections between the fate of $RO_2$, HOM forming mechanisms, and camphene SOA formation.

## 2 Methods

### 2.1 Chamber Facility and Instrumentation

The camphene photooxidation experiments were conducted in the University of California, Riverside (UCR) dual indoor environmental chamber. Chamber characterization and features have been previously described in detail (Carter et al., 2005). Briefly, the UCR environmental chamber consists of two 90 $m^3$ collapsible Teflon reactors (2MIL (0.0508 mm) FEP film) kept at a positive pressure differential (3.73–4.98 Pa) to the enclosure where the reactors are located to minimize contamination during experiments. The enclosure is relative humidity controlled (<0.1%), temperature controlled (300 ± 1 K), and continuously flushed with dry purified air (dew point < -40 °C). Prior to and between experiments, reactors were collapsed to a volume < 20 $m^3$ for cleaning. The cycle of filling-purging the reactors was repeated until particle number concentrations were < 5 $cm^{-3}$ and $NO_x$ mixing ratios were < 1 ppb. The reactors were then flushed with dry purified air and filled up to 90 $m^3$ overnight. The filling-purging of the reactors is controlled by an "elevator" program in LabView.

NO and $NO_2$ mixing ratios were monitored by a Thermo Environmental Instruments Model 42C chemiluminescence $NO_x$ analyzer. $O_3$ mixing ratios were monitored by a Dasibi Environmental Corp. Model 1003-AH $O_3$ analyzer. An Agilent 6890 gas chromatograph with flame ionization detector (GC-FID) was used to measure the camphene levels during the experiments.

Multiple instruments were used for particle-phase monitoring. Each reactor was equipped with a scanning mobility particle sizer (SMPS), including a TSI 3081 differential mobility analyzer (DMA), to measure the particle mass concentration. Particle effective density was directly measured by an Aerosol Particle Mass Analyzer (APM, Kanomax) with a SMPS built in house (Malloy et al., 2009). Chemical composition of SOA was measured using HR-ToF-AMS (DeCarlo et al., 2006) and analyzed to obtain O:C and H:C ratios by applying the method of Canagaratna et al. (2015). Data processing was performed using the ToF-AMS Analysis Toolkit 1.57 and PIKA 1.16 on Igor Pro 6.36. A prior characterization of this UCR chamber system (Li et al., 2016) reported an experimental uncertainty in SOA yields of < 6.65%.

Particle wall loss corrections were performed following the method described in Cocker et al. (2001). Vapor wall loss of organics has been reported in multiple chambers (e.g., Zhang et al., 2015, 2014; Schwantes et al., 2019); and has been modeled as a function of the mass and volatility of the condensing compounds, condensation sink, and characteristics of the chamber (e.g., La et al., 2016; Zhang et al., 2014; Ye et al., 2016). The extent to which these observations and modeling simulations are relevant in the UCR chamber is unclear, given the significant difference in chamber sizes. The UCR chamber is 4.5 times larger (90 $m^3$) than the largest referenced chamber in these studies (20 $m^3$) and most are ~10 $m^3$. In the UCR chamber, the role of vapor wall loss has been investigated in SOA

experiments using various precursor compounds (including α-pinene and *m*-xylene) under seed and no seed conditions (Clark et al., 2016; L. Li et al., 2015). There has been no evidence of non-negligible vapor wall loss in those experiments, including no measurable differences in SOA formation in experiments with and without seed. In this work, stability tests on camphene demonstrated negligible vapor wall loss of the parent compound. Thus without evidence to suggest otherwise, negligible vapor wall loss was assumed for these experiments. This assumption is further discussed where it may affect the major conclusions regarding the role of gas-phase chemistry on SOA formation.

## 2.2 Experimental Conditions

A series of 13 camphene photooxidation experiments were carried out under varying levels of camphene and $NO_x$ (Table 1). Due to the relatively high melting point of camphene (51 °C), camphene (Sigma-Aldrich, purity > 96 %, FG) was injected into a glass manifold (heated to 50 °C by heating tape) using a preheated (~50-55 °C) microliter syringe. As camphene evaporated it was carried to the reactors by dry purified compressed air flowing through a glass manifold at 8 LPM for 15 mins. Injection lines from the glass manifold to the reactors were also heated to reduce losses of camphene. $H_2O_2$ (Sigma Aldrich, 50 wt.% in $H_2O$) was injected by adding 200 μl onto glass wool in glass tubing and then placing the tubing in a 56 °C oven with 10 LPM of dry purified compressed air flowing through the tubing for 15 mins and into the reactors. An inert tracer, perfluorohexane (Sigma-Aldrich, 99 %) or perfluorobutane (Sigma-Aldrich, 99 %), was injected to the reactors through the heated glass manifold by a carrier gas of 50 °C pure $N_2$. NO (Matheson, UHP) at known volume and pressure was transferred and injected through the same glass manifold as the inert tracer. When gaseous injection of camphene, $H_2O_2$, inert tracer, and NO (when used) was completed, the reactors were internally mixed using built-in blowers to ensure uniform distribution of chemicals, and then irradiated using UV black lights (115w Sylvania 350BL) to start photooxidation. No seed aerosol was used in this study. All experiments were conducted under dry conditions (relative humidity < 0.1 %) at 300 K. The initial conditions of the experiments are summarized in Table 1.

**Table 1**. Summary of initial conditions for chamber experiments and box model simulations.

| | Expt. | Initial Conditions for Chamber Experiments and SAPRC Simulations | | | | Initial Conditions for GECKO-A Simulations | | | |
| --- | --- | --- | --- | --- | --- | --- | --- | --- | --- |
| | | Camphene (ppb) | Added NOx (ppb) | *H2O2 (ppb) | HC/NOx (ppb/ppb) | Camphene (ppb) | NOx (ppb) | H2O2 (ppb) | HC/NOx (ppb/ppb) |
| without NOx | WO1 | 7 | | 854 | | 10 | | 1000 | |
| | WO2 | 9 | | 1148 | | | | | |
| | WO3 | 28 | | 1212 | | 25 | | 1000 | |
| | WO4 | 57 | | 1182 | | 50 | | 1000 | |
| | WO5 | 120 | | 1212 | | 100 | | 1000 | |
| | WO6 | 223 | | 1576 | | 150 | | 1000 | |
| with NOx | W1 | 7 | 89 | 854 | 0.08 | 10 | 80 | 1000 | 0.13 |
| | W2 | 25 | 138 | 1040 | 0.18 | 25 | 80 | 1000 | 0.31 |
| | W3 | 32 | 62 | 1136 | 0.51 | | | | |
| | W4 | 43 | 7 | 860 | 5.91 | 50 | 80 | 1000 | 0.63 |
| | W5 | 60 | 94 | 1227 | 0.64 | | | | |
| | W6 | 131 | 98 | 1167 | 1.33 | 100 | 80 | 1000 | 1.25 |
| | W7 | 172 | 60 | 1121 | 2.88 | 150 | 80 | 1000 | 1.88 |

* $H_2O_2$ mixing ratio was targeted at 1ppm but corrected based on tracer (perfluorohexane or perfluorobutane) concentration to offset initial reactor volume bias. Corrected $H_2O_2$ mixing ratios were used in SAPRC modeling.

### 2.3  Model Configurations and Conditions

The chamber experiments were modeled using two different box models, SAPRC and GECKO-A. The SAPRC model was chosen because it has been designed to evaluate gas-phase chemistry in the UCR chamber. The GECKO-A model was chosen because of the ability to predict both gas and particle phase composition, and the prior work of Afreh et al. (2020), in which GECKO-A was used to study SOA formation from camphene. The initial conditions of the simulations are summarized in Table 1.

### 2.3.1 SAPRC

A gas-phase oxidation mechanism was derived using the SAPRC mechanism generation system (MechGen) with modified initial rate constants (camphene with OH, $NO_3$ and $O_3$) based on published literature data (Atkinson and Arey, 2003). MechGen, described elsewhere (Carter, 2021; Carter, 2020b; Jiang et al., 2020), is capable of generating fully explicit mechanisms for the atmospheric reactions of many types of organic compounds and the intermediate radicals they form. MechGen uses experimentally derived rate constants and branching ratios if data are available and otherwise uses estimated rate constants and branching ratios based on group additivity and other estimation methods. This system was used to derive reactions of explicit and lumped organic compounds and products in the development of the SAPRC-18 mechanism (Carter, 2020a) and a detailed SAPRC furans mechanism (Jiang et al., 2020).

The MechGen-derived camphene mechanism was implemented into the SAPRC box model to simulate chamber experiments under the same chemical conditions as the chamber experiments, where the initial hydrocarbon concentrations and $NO_x$ levels were as defined in Table 1. The SAPRC box model system has been used for chemical

mechanism development, evaluation, and box modeling applications since the mid-1970s (Carter, 1990, 1994, 2000, 2010a, 2010b, 2020a). The initial conditions and relevant chemical parameters for environmental chamber experiments are required inputs; simulations can be performed using multiple versions of the SAPRC gas-phase chemical mechanism. In this work, the recently published version, SAPRC-18 (Carter, 2020a), was selected as the base mechanism because it represents the current state of the science and includes the most up-to-date model species and explicit representation of $RO_2$ chemistry.

### 2.3.2 GECKO-A

GECKO-A is a nearly explicit mechanism generator and SOA box model. GECKO-A relies on experimental data and structure-activity relationships (SARs) to generate detailed oxidation reaction schemes for organic compounds. The generated reaction schemes are applied in the SOA box model to simulate SOA formation based on the absorptive gas/particle partitioning model of Pankow (1994), where thermodynamic equilibrium between the gas and an ideal particle phase is assumed. Detailed descriptions of GECKO-A, including mechanism generation and SOA formation, are provided by Aumont et al. (2005) and Camredon et al. (2007). GECKO-A has been used to predict SOA in a number of studies (e.g., Aumont et al., 2012; Lannuque et al., 2018; McVay et al., 2016), including camphene (Afreh et al., 2020). Details of the camphene mechanism and SOA box modeling were described in Afreh et al. (2020). Briefly, the camphene mechanism includes $1.3 \times 10^6$ reactions and $1.8 \times 10^5$ oxidation products; vapor pressures of products were calculated based on the Nannoolal method (Nannoolal et al., 2008).

The GECKO-A simulations were performed for a predefined set of conditions, prior to the chamber experiments, and thus in some cases differ from the experimental conditions. GECKO-A simulations were performed under two $NO_x$ conditions, with 80 ppb of $NO_x$ and without $NO_x$ (Table 1). For both $NO_x$ conditions, the initial hydrocarbon mixing ratios were set at 10, 25, 50, 100, and 150 ppb. All simulations were run under the following initial conditions: 1000 ppb of $H_2O_2$, 1 $\mu g\ m^{-3}$ of organic seed with molecular weight of 250 g $mol^{-1}$, 298 K temperature, 1% relative humidity, and $50°$ solar zenith angle (required to compute the photolysis frequencies). Simulation results for camphene were compared with chamber data including SOA mass yields, precursor decay rates, and oxidant levels.

### 3. Experimental and Modeling Results

Table 2 summarizes the measured initial $NO/NO_2$ mixing ratios, initial camphene concentration ($[HC]_0$), reacted camphene concentration ($\Delta[HC]$), SOA mass ($M_o$) formed, particle density, final peak particle diameter ($d_p$), photochemical aging time, irradiation time, and SOA mass yield (SOA mass formed, $M_o$/hydrocarbon reacted, $\Delta HC$) for all 13 experiments. Except for Fig. 4, in which SOA mass yields are shown as a function of photochemical age, all SOA mass yields refer to the mass at the end of the experiments (~6 hours). Measured and predicted gas-phase species are presented in Sect. 3.1; SOA mass and yields are presented in Sect. 3.2. The predicted fate of $RO_2$ in the context of initial HC to initial $NO_x$ mixing ratio ($[HC]_0/[NO_x]_0$) is presented in Sect. 3.3.

**Table 2.** Chamber SOA data, WO indicates experiments without added $NO_x$ and W with added $NO_x$.

| Expt. | Initial NO/NO$_2$ | [HC]$_0$ | $\Delta$[HC] | $M_o$ | PM den. | **Peak $d_p$ | Irradiation time | Photochemical aging time | SOA mass yield |
|---|---|---|---|---|---|---|---|---|---|
| | ppb | µg m$^{-3}$ | µg m$^{-3}$ | µg m$^{-3}$ | g cm$^{-3}$ | nm | hour | hour | |
| WO1 | 0/0 | 41 | 41 | 6.1 | 1.42 | 126 | 4.9 | 16.1 | 0.15 |
| WO2 | 0/0 | 49 | 49 | 3.7 | 1.42 | 125 | 5.0 | 16.7 | 0.08 |
| WO3 | 0/0 | 155 | 153 | 42.0 | *1.36 | 214 | 6.1 | 17.7 | 0.27 |
| WO4 | 0/0 | 313 | 305 | 84.4 | *1.34 | 270 | 6.7 | 15.8 | 0.28 |
| WO5 | 0/0 | 663 | 597 | 158.6 | 1.30 | 286 | 6.7 | 9.5 | 0.27 |
| WO6 | 0/0 | 1230 | 844 | 162.4 | *1.31 | 492 | 6.1 | 5.0 | 0.19 |
| W1 | 86/2 | 40 | 40 | 14.6 | 1.46 | 120 | 5.1 | 50.6 | 0.36 |
| W2 | 114/24 | 140 | 140 | 46.1 | 1.47 | 188 | 5.2 | 40.6 | 0.33 |
| W3 | 51/11 | 177 | 177 | 112.3 | *1.44 | 185 | 6.0 | 42.0 | 0.64 |
| W4 | 5/2 | 238 | 237 | 96.0 | 1.35 | 290 | 5.9 | 16.1 | 0.41 |
| W5 | 45/49 | 334 | 334 | 199.5 | *1.44 | 430 | 5.8 | 33.6 | 0.60 |
| W6 | 42/56 | 724 | 724 | 428.8 | *1.42 | 665 | 5.8 | 12.7 | 0.59 |
| W7 | 45/15 | 956 | 950 | 494.3 | *1.39 | 800 | 6.4 | 8.75 | 0.52 |

* Estimated using best fit line shown in Fig. S6.
** Peak $d_p$ refers to the diameter of particles at the peak of the size distribution plot at the end of the experiment. The uncertainty of peak $d_p$ values is less than 5%.

### 3.1 Gas-Phase Reactivity

Figure 2 shows measured and predicted camphene consumption for the 13 photooxidation experiments, and the calculated time-dependent $\beta$ values (ratio of RO$_2$ + NO to the sum of RO$_2$ + NO and RO$_2$ + HO$_2$) (Henze et al., 2008; Pye et al., 2010) based on SAPRC predictions for each experimental condition. Additional comparisons of measured and predicted gas-phase species are shown in Fig. S1. Higher camphene decay rates and higher OH levels (0.15–0.88 ppt with added $NO_x$; 0.05–0.29 ppt without added $NO_x$) were observed and predicted for experiments with added $NO_x$ than without; likely due to the fast recycling of OH when $NO_x$ was present (Fig. 2). For all experiments, the $\beta$ values changed as a function of time due to changing chemical conditions. Note that due to off-gassing of $NO_x$ from the Teflon reactor (Carter et al., 2005), $\beta$ values simulated here were larger than 0 even for experiments without added $NO_x$. Experiments with added $NO_x$ have $\beta$ values from 0.12–1, while experiments without added $NO_x$ have values < 0.12. For all parameters (camphene consumption, $NO_x$ decay, O$_3$ formation, and OH levels), the SAPRC simulation results were generally in good agreement with the experimental data. The exception to the generally good agreement was O$_3$ predictions in experiments without added $NO_x$, which have a relatively strong dependence on the HONO off-gassing rate. The quantity $\Delta$([O$_3$]-[NO]) has been used to evaluate the rate of NO oxidation by RO$_2$ for VOC-$NO_x$ systems in SAPRC mechanism development (Carter and Lurmann, 1990; Carter, 1999; Carter, 2009; Carter, 2020). Figure S2 shows the comparison of the $\Delta$([O$_3$]-[NO]) values between chamber measurements and SAPRC simulations for experiments with added $NO_x$. The SAPRC box model captures the rates of RO$_2$+NO well, and supports the use of

the SAPRC model to interpret chamber observations especially in the presence of $NO_x$. Unfortunately, it is hard to quantify how well constrained the other $RO_2$ reaction rates and product yields are without corresponding measurements, which are not available. In this case, the SAPRC model was largely used to probe the mechanism (diagnostic) and not to predict yields (prognostic).

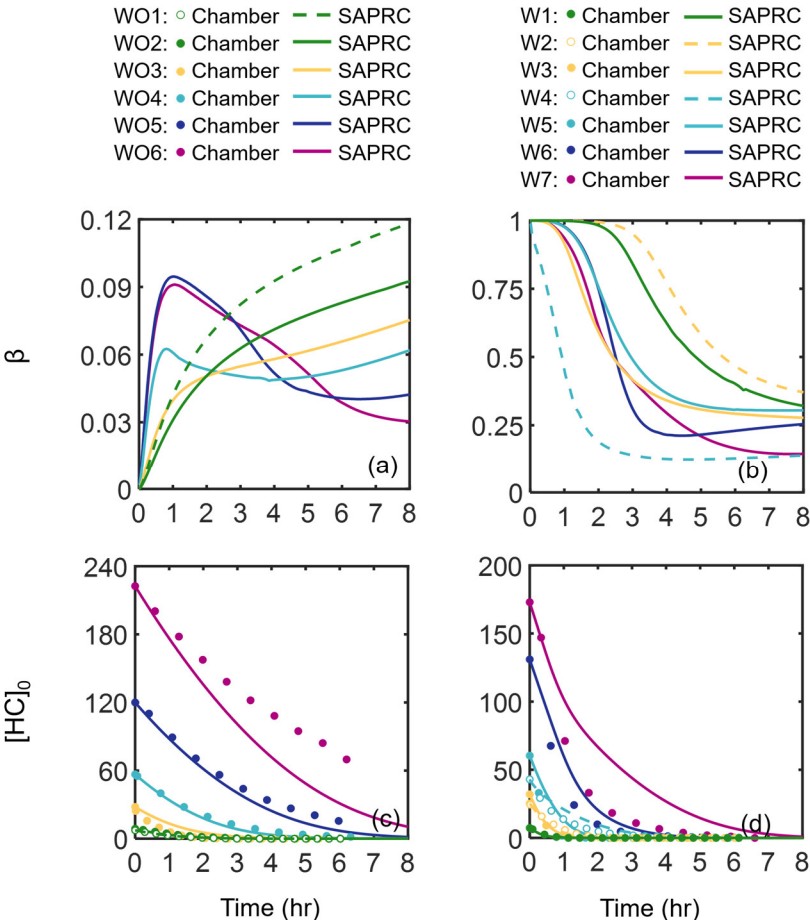

**Figure 2. SAPRC predicted $\beta$ values: (a) without added $NO_x$, and (b) with added $NO_x$. Measured (circles) and predicted (lines) camphene consumption as a function of irradiation time: (c) without added $NO_x$, and (d) with added $NO_x$. The hollow makers used in (c) and (d) are equivalent to dashed lines defined in the legends.**

**3.2 SOA Mass and Yield**

Measured SOA mass yields are shown in Fig. 3 as a function of SOA mass ($M_o$) for experiments with (squares) and without (circles) added $NO_x$. The SOA mass yields were much higher in experiments with added $NO_x$ (0.33–0.64) than experiments without added $NO_x$ (0.08–0.28). The SOA mass yields measured at the lowest $[HC]_0/ \Delta[HC]$ may be an underestimate due to the assumption of negligible vapor wall loss, which would have the largest effect at low $\Delta[HC]$ (Krechmer et al., 2020). The observed trends in SOA mass yields were unexpected based on prior chamber studies of SOA formation from monoterpenes, such as OH oxidation studies of α- and β-pinene, in which SOA mass

yields were reported to be suppressed under high-$NO_x$ conditions (Eddingsaas et al., 2012; Pullinen et al., 2020; Sarrafzadeh et al., 2016).

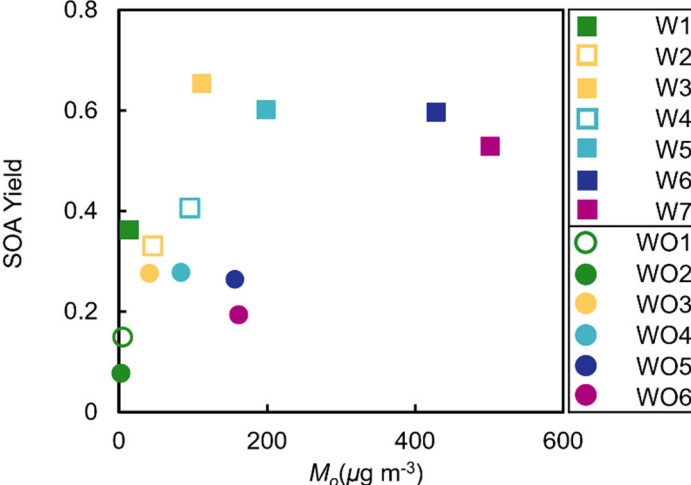

**Figure 3. Measured camphene SOA mass yields as a function of SOA mass ($M_o$). Squares indicate experiments with (W) and circles without (WO) added $NO_x$. Initial HC mixing ratios are differentiated by color; open symbols are used to indicate replicate initial HC mixing ratios.**

Figure 3 shows another unexpected observation: the SOA mass yields decreased at high SOA mass under both $NO_x$ conditions. In the presence of $NO_x$, the observed SOA mass yields increased with $M_o$ for $M_o \leq 112$ µg m$^{-3}$, plateaued between 112 µg m$^{-3}$ < $M_o \leq 429$ µg m$^{-3}$, and then decreased for $M_o > 429$ µg m$^{-3}$. Without $NO_x$, the observed SOA mass yields increased for $M_o \leq 42$ µg m$^{-3}$, plateaued between 42 µg m$^{-3}$ < $M_o \leq 159$ µg m$^{-3}$, and then decreased for $M_o > 159$ µg m$^{-3}$. The difference between the peak SOA mass yield and the SOA mass yield at the highest [HC]$_0$ was ~0.12 with added $NO_x$ and ~0.08 without added $NO_x$. While the SOA mass yields at the highest [HC]$_0$ may not be statistically different within the uncertainty of the measurements, this trend was also observed in the GECKO-A model simulations (see Sect. 5) and thus were further investigated, and reasonably explained, by the $RO_2$ fate based on box model simulations (see Sect. 4 & 5).

The varying [OH] levels in the chamber experiments led to a wide range of photochemical aging times, from hours to days. The irradiation time was converted to equivalent photochemical aging time in the ambient atmosphere using equation (1) (Aumont et al., 2012):

$$\tau = \frac{1}{[OH]_{atm}} \int_0^t [OH]_{sim} dt \tag{1}$$

where [OH]$_{atm}$ was assumed to be $2 \times 10^6$ molecule cm$^{-3}$. Figure 4 shows the measured SOA mass yields as a function of photochemical aging time calculated using OH values predicted by SAPRC ([OH]$_{sim}$). The SOA mass yields are dependent on OH levels and thus photochemical aging time. The yield curves for most experiments plateaued or nearly plateaued by the end of the experiment. Higher [HC]$_0$ generally led to steeper increases in SOA mass yield as a function of aging time. Experiments with added $NO_x$ generally had longer photochemical aging times than experiments without added $NO_x$; without added $NO_x$, all experiments may not have fully plateaued and thus would

have had higher SOA mass yields at longer irradiation times. However, even at the same aging time (Fig. S8), the SOA yields were higher in the experiments with added $NO_x$. The higher SOA mass yields in experiments with added $NO_x$ may partially be attributed to the difference in [OH] levels and extents of aging. Similar $NO_x$ effects have been reported in many previous studies (e.g., Ng et al., 2007a; Sarrafzadeh et al., 2016). Sarrafzadeh et al. (2016) proposed that in a study of β-pinene the OH level was the main factor that accounted for differences in SOA mass yields under varying $[NO_x]_0$. In the camphene experiments presented herein, the aging effects were determined to be less important than $RO_2$ chemistry, since the SOA mass yield curves as a function of photochemical aging already plateau or nearly plateau by the end of experiments (Fig. 4) and the shapes of the growth curves (Fig. S9 and Fig. S10) indicate different kinetics and contributions from oxidation products that form slowly among and between experiments with and without added $NO_x$ (Ng et al., 2006).

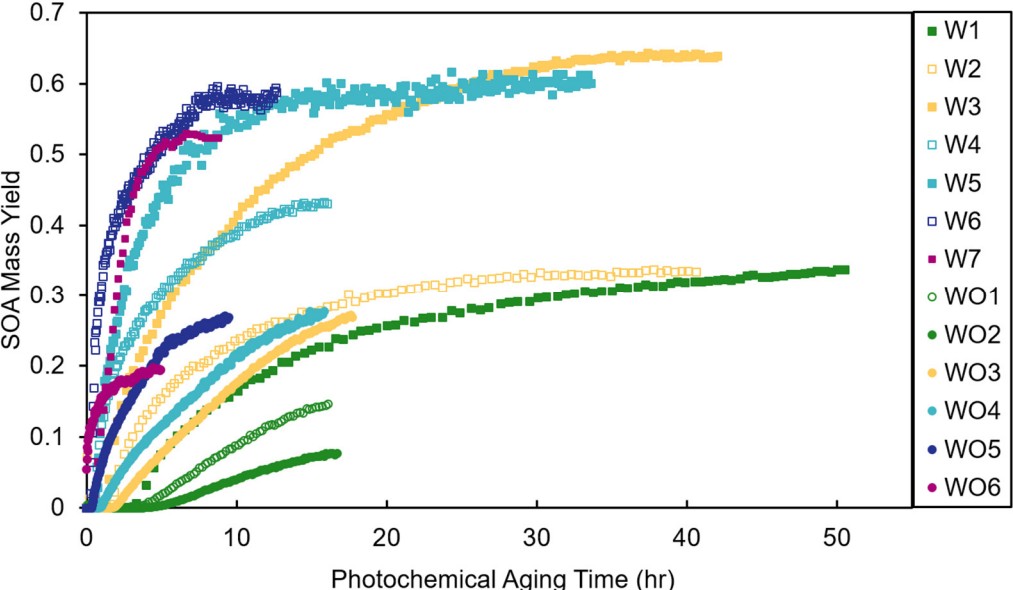

**Figure 4. Measured SOA mass yields as a function of photochemical aging time in experiments with added $NO_x$ (squares) and experiments without added $NO_x$ (circles).**

SOA mass yields are shown as a function of Δ[HC], $[HC]_0/[NO_x]_0$, and photochemical aging time in Fig. 5. For the experiments without added $NO_x$, a constant value of 1 ppb was used in the calculations of $[HC]_0/[NO_x]_0$ to account for $NO_x$ off-gassing from the Teflon reactors. Based on recent characterization experiments, the UCR chamber has a $NO_x$ off-gassing rate of 2.8 ppt/min in the form of HONO; the camphene experiments lasted for ~300 to 360 mins. Over low Δ[HC], SOA mass increased in experiments without added $NO_x$ due to the dependence of partitioning on $M_o$ (or Δ[HC]). This trend may be exaggerated due to the assumption of negligible vapor wall loss, which could result in an underestimation of SOA mass yield particularly at low Δ[HC] (Krechmer et al., 2020). The sensitivity of SOA formation to $[HC]_0/[NO_x]_0$ over the range of [HC] sampled is also shown. At a given Δ[HC] level, a lower $[HC]_0/[NO_x]_0$ (when within 0.5–200) led to a higher SOA mass yield (decreasing $[HC]_0/[NO_x]_0$ by ~2 orders of magnitude resulted in a factor of two increase in SOA mass yield). The chamber data presented here indicate that the highest SOA mass yields from camphene were observed in a regime of high Δ[HC] and moderate $[HC]_0/[NO_x]_0$; this

regime is distinguished from an extreme [NOx] regime, proposed in section 4.2, in which SOA mass yields are suppressed at the lowest $[HC]_0/[NO_x]_0$ (also shown in Fig. 5). These observations are different from those in studies of α-pinene, in which lower $[HC]_0/[NO_x]_0$ generally led to lower SOA mass yield (Eddingsaas et al., 2012). The observed trends are further explored in the following sections, particularly the role of $RO_2$ based on SAPRC simulations.

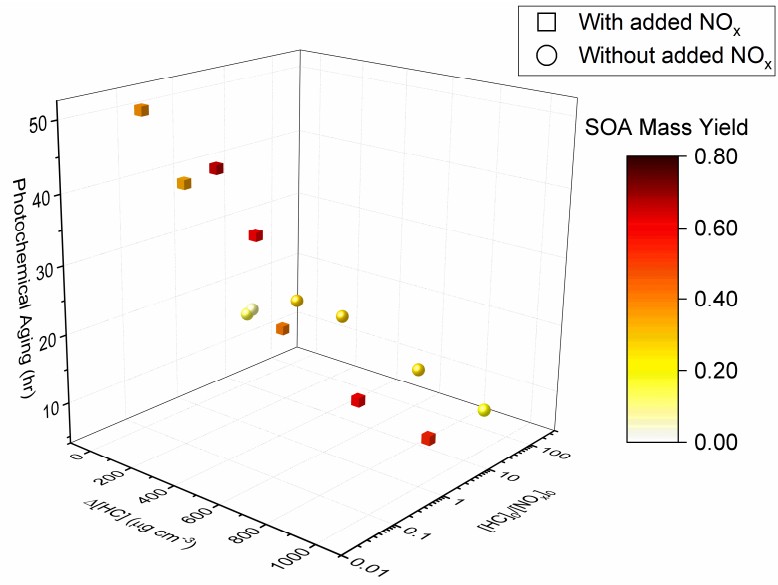


**Figure 5. SOA mass yield (color bar) as a function of Δ[HC], $[HC]_0/[NO_x]_0$, and photochemical aging time, with added $NO_x$ experiments square markers and without added $NO_x$ experiments round markers.**

### 3.3 $[HC]_0/[NO_x]_0$ and the Fate of Peroxy Radicals

Table S1 shows the experimental $[HC]_0/[NO_x]_0$ and the SAPRC predicted fate of total $RO_2$ (calculated as the

summation of $RO_2$ radicals that undergo bimolecular reactions) for all the chamber runs. In Fig. 6, the fate of total $RO_2$ is shown as a function of $[HC]_0/[NO_x]_0$. The majority of $RO_2$ was predicted to undergo bimolecular reactions with $HO_2$ or NO across the range of $[HC]_0/[NO_x]_0$ values sampled. At $[HC]_0/[NO_x]_0 < 6$, > 50% of the $RO_2$ was predicted to react with NO; and at $[HC]_0/[NO_x]_0 > 10$, > 50% of the $RO_2$ was predicted to react with $HO_2$. A roughly 50:50 branching of $RO_2$ between NO and $HO_2$ was reached when $[HC]_0/[NO_x]_0$ was 6:1, which is close to the ratio

that was suggested in Presto et al. (2005). When $[HC]_0/[NO_x]_0$ increased over 50, the total fraction of bimolecular $RO_2$ + $RO_2$ increased from 0 to 30%. In addition, the normalized total $RO_2$ concentration (total $[RO_2]/[HC]_0$, ppbv/ppbv) increased as $[HC]_0/[NO_x]_0$ decreased (Fig. 7), suggesting more oxygenated $RO_2$s were formed by NO pathway than others, which is consistent with the formation of HOMs with added $NO_x$. There is a general trend of increasing SOA mass yield with decreasing $[HC]_0/[NO_x]_0$ (Fig. 5 and Fig. 7), with the exception of four outliers (W1, W2, WO1, and

WO2) that have relatively low SOA mass yields. Experiments WO1, WO2, W1 had the lowest Δ[HC] (49, 41, and 40 μg/m³, respectively, Table 2), indicating the SOA mass yields were influenced by Δ[HC] as well as $RO_2$ chemistry. The connections between the fate of $RO_2$ and observed SOA mass yields are further discussed in Sect. 4. Though vapor wall loss has been found to be negligible in previous UCR chamber experiments, such experiments were

typically conducted at higher $[HC]_0$. Thus, it is acknowledged that vapor wall loss could affect the measured SOA

yields, particularly for experiments W1-2 and WO1-2 with low $[HC]_0$ (or $M_o$). A vapor wall loss correction for those

experiments would increase the measured SOA, but would not affect the following discussion or conclusions regarding

the role of $RO_2$ chemistry.

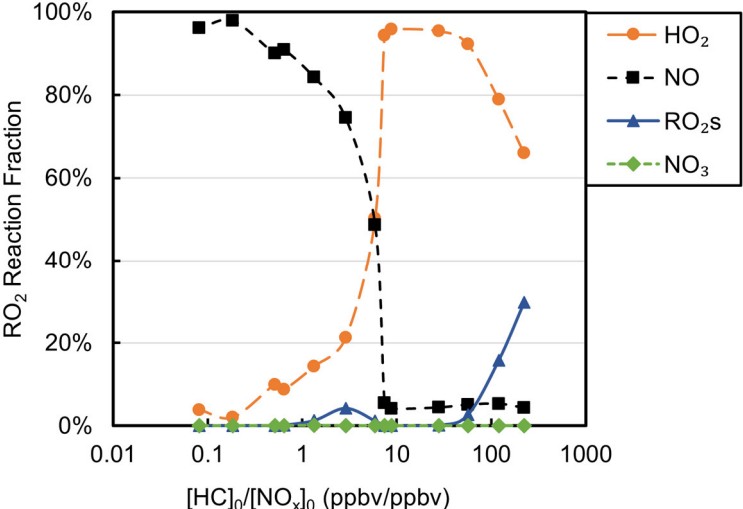

**Figure 6. Fractions of total $RO_2$ reactions of each type as a function of $[HC]_0/[NO_x]_0$ based on Table S1.**

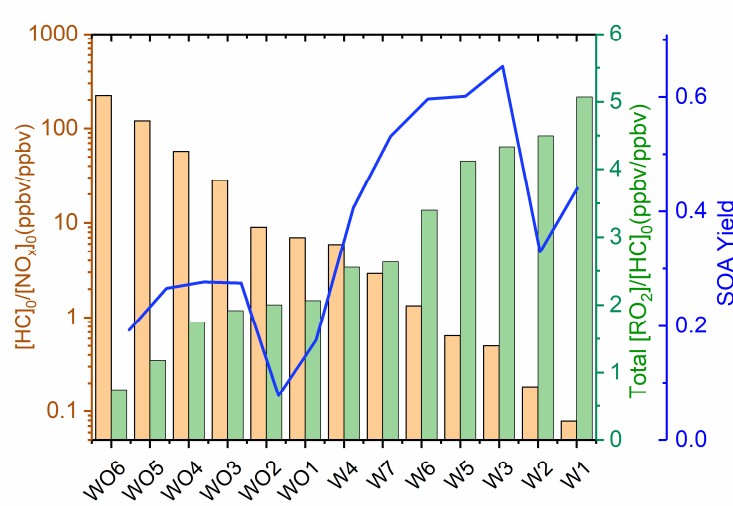


**Figure 7. Relationship between total $[RO_2]/[HC]_0$, $[HC]_0/[NO_x]_0$, and SOA mass yields.**

By assuming the gas-phase chemistry and product distribution were similar when $RO_2 + NO$ accounted for more

than 80% of the total $RO_2$ consumption and when $RO_2 + HO_2$ accounted for more than 80% of the total $RO_2$

consumption, experiments with (W1–3, 5–6) and without (WO1–4) added $NO_x$ were grouped and used to derive SOA

parameters using the two-product (Odum et al., 1996) and VBS approaches (Donahue et al., 2006; Donahue et al.,

2009). The resultant parameters are shown in Table 3 (two-product) and Table 4 (VBS).

**Table 3.** Two-Product Model SOA parameters.

| | $\alpha_1$ | $\log_{10} C^*_1$ | $\alpha_2$ | $\log_{10} C^*_2$ |
|---|---|---|---|---|
| Without NO$_x$ | 0.0017 | 1.08 | 0.3139 | 0.92 |
| With NO$_x$ | 0.4484 | 1.77 | 0.2398 | -2.94 |

**Table 4.** VBS Model SOA parameters.

| $C^*$ | $^\dagger\alpha_{wo}$ | $^\dagger\alpha_w$ |
|---|---|---|
| 0.1 | 0.0001 | 0.2657 |
| 1 | 0.0152 | 0.0008 |
| 10 | 0.3069 | 0.0357 |
| 100 | 0.0001 | 0.4222 |
| 1000 | 0.0003 | 0.0000 |

$^\dagger$ wo refers to without added NO$_x$; w refers to with added NO$_x$.

## 4 Discussion

The reaction rate constant of camphene with O$_3$ is relatively low compared to OH, and thus it is expected that OH is the dominant oxidant in the photooxidation of camphene under chamber conditions, especially with the high initial H$_2$O$_2$ (~1 ppm) concentrations. This is supported by SAPRC simulation results (see Fig. S3 in SI), in which O$_3$ accounts for 0–3% and NO$_3$ for 0–16% of camphene oxidation, demonstrating the important role of OH oxidation in these studies.

### 4.1 Camphene + OH Gas-phase Mechanism

Figure 8 shows the MechGen predicted reactions and products of OH-initiated oxidation of camphene in the presence of NO$_x$ through one major pathway, which had a yield of 0.83 (a more detailed reaction mechanism schematic is presented in Fig. S4). The reaction starts with OH addition to the CH$_2$=(C) position to form a ring-retaining alkyl radical, which further reacts with O$_2$ to form the camphene peroxy radical, RO$_2$-a. RO$_2$-a can react with oxidants (NO, NO$_3$, HO$_2$, and/or other RO$_2$) to create an alkoxy radical, RO-a, with NO to NO$_2$ conversion; or form stable products such as organic nitrate (NO3CAMP1), hydroperoxide (HO2CAMP1), and alcohol (RO2CAMP1) compounds. The cyclic alkoxy radical RO-a can undergo prompt beta (β)-scission ring-opening reaction, and then O$_2$ addition to form another peroxy radical, RO$_2$-b. In the presence of NO$_x$, rapid β-scission decomposition, or ring-opening reactions of the camphene alkoxy radicals (RO-b and RO-c) occur through the RO$_2$ + NO pathway, leading to the generation of the peroxy radical RO$_2$-d with lower carbon number and higher O:C ratio (increases from 0.30 for RO$_2$-a to 0.71 for RO$_2$-d).

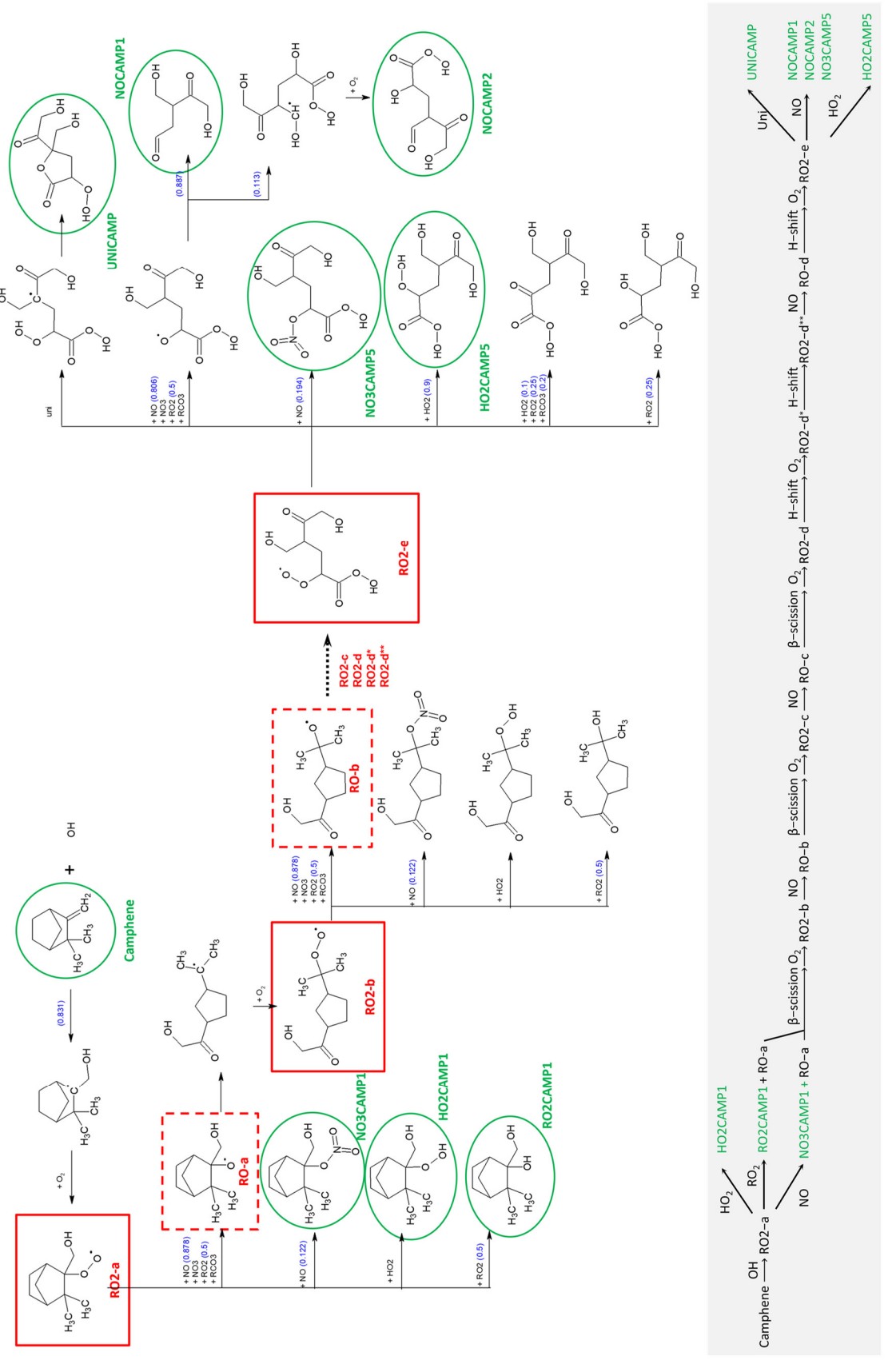

**Figure 8. Schematic of the OH-initiated oxidation of camphene mechanism in SAPRC at 298 K and atmospheric pressure in the presence of NO$_x$. Check Fig. S4 for more details.**

MechGen predicted that RO$_2$-d could undergo 1,5 H-shift isomerization nearly instantaneously, even in the presence of ~ 100 ppb NO$_x$. Subsequent rapid addition of O$_2$ can form a new peroxy radical RO$_2$-d* which can undergo 1,7 H-shift isomerization and form the peroxy radical RO$_2$-d$^{**}$. RO$_2$-d$^{**}$ can participate in termination reactions with NO and HO$_2$ to form organic nitrate (NO3CAMP4) and hydroperoxide (HO2CAMP4) products, which are known as highly oxygenated organic molecules (HOMs). In the presence of NO$_x$, RO$_2$-d$^{**}$ can also react with NO to form the

alkoxy radical RO-d that can undergo 1,4 H-shift isomerization and then O$_2$ addition to form the new peroxy radical RO$_2$-e which will also lead to the formation of HOMs such as NO3CAMP5, HO2CAMP5, and UNICAMP. A recent SOA study by Mehra et al. (2020) demonstrated the formation of HOMs in camphene chamber experiments under both low NO$_x$ (30 ppb camphene, ~ 0 ppb NO$_x$) and medium NO$_x$ (30 ppb camphene, 2.2 ppb NO, 58.4 ppb NO$_2$) conditions. Based on their observations and analysis, the average molecular formula of the camphene SOA was

C$_{7.26}$H$_{9.85}$O$_{4.03}$ for low NO$_x$ and C$_{6.63}$H$_{9.7}$N$_{0.12}$O$_{4.21}$ for the medium NO$_x$ conditions, which also suggest the occurrence of ring-opening and decomposition reactions during camphene photooxidation, as predicted by MechGen.

### 4.2 The Formation of HOMs and Influence on SOA Mass Yields

**Table 5**. Log$_{10}$ $C^*$ value for selected 1$^{st}$ generation of stable end products formed from camphene reactions with OH.

| Species | Atom | | | | O:C | log$_{10}$ $C^*$ | Species | Atom | | | | O:C | log$_{10}$ $C^*$ |
|---|---|---|---|---|---|---|---|---|---|---|---|---|---|
| | C | H | O | N | | | | C | H | O | N | | |
| HO2CAMP1 | 10 | 18 | 3 | 0 | 0.30 | 2.5 | NO3CAMP1 | 10 | 17 | 4 | 1 | 0.40 | 3.5 |
| HO2CAMP2 | 10 | 18 | 4 | 0 | 0.40 | 1.7 | NO3CAMP2 | 10 | 17 | 5 | 1 | 0.50 | 2.6 |
| HO2CAMP3 | 7 | 12 | 4 | 0 | 0.57 | 2.5 | NO3CAMP3 | 7 | 11 | 5 | 1 | 0.71 | 3.5 |
| HO2CAMP4 | 7 | 12 | 7 | 0 | 1.00 | -1.3 | NO3CAMP4 | 7 | 11 | 8 | 1 | 1.14 | -0.1 |
| HO2CAMP5 | 7 | 12 | 8 | 0 | 1.14 | -4.3 | NO3CAMP5 | 7 | 11 | 9 | 1 | 1.29 | -2.8 |
| RO2CAMP1 | 10 | 18 | 2 | 0 | 0.20 | 3.8 | NOCAMP1 | 6 | 10 | 4 | 0 | 0.67 | 2.6 |
| UNICAMP | 7 | 10 | 7 | 0 | 1.00 | -3.9 | NOCAMP2 | 7 | 10 | 7 | 0 | 1.00 | -1.1 |


**Table 6**. Fractions of peroxy radical $RO_2$-a reactions of each type, calculated based on SAPRC simulations.

| Expt. | $[HC]_0$ (ppb) | *$[HC]_0/[NO_x]_0$ (ppbv/ppbv) | SOA Mass Yield | Fraction of $RO_2$-a Reaction | | | | |
|---|---|---|---|---|---|---|---|---|
| | | | | NO | $HO_2$ | $RCO_3$ | $RO_2$ | $NO_3$ |
| WO1 | 7 | 7 | 0.15 | 0.03 | 0.97 | 0 | 0 | 0 |
| WO2 | 9 | 9 | 0.08 | 0.02 | 0.98 | 0 | 0 | 0 |
| WO3 | 28 | 28 | 0.27 | 0.02 | 0.97 | 0 | 0 | 0 |
| WO4 | 57 | 57 | 0.28 | 0.03 | 0.89 | 0 | 0.08 | 0 |
| WO5 | 120 | 120 | 0.27 | 0.03 | 0.64 | 0.02 | 0.30 | 0 |
| WO6 | 223 | 223 | 0.19 | 0.03 | 0.54 | 0.02 | 0.41 | 0 |
| W1 | 7 | 0.08 | 0.36 | 1.00 | 0 | 0 | 0 | 0 |
| W2 | 25 | 0.18 | 0.33 | 1.00 | 0 | 0 | 0 | 0 |
| W3 | 32 | 0.51 | 0.64 | 0.97 | 0.03 | 0 | 0 | 0 |
| W4 | 43 | 5.91 | 0.41 | 0.46 | 0.53 | 0.01 | 0 | 0 |
| W5 | 60 | 0.64 | 0.60 | 0.97 | 0.03 | 0 | 0 | 0 |
| W6 | 131 | 1.33 | 0.59 | 0.88 | 0.12 | 0.01 | 0.01 | 0 |
| W7 | 172 | 2.88 | 0.52 | 0.65 | 0.30 | 0.03 | 0.01 | 0 |

*The $[HC]_0/[NO_x]_0$ for WO1–6 experiments were estimated assuming 1 ppb of $NO_x$.

Table 5 lists the log $C^*$ values and O:C ratios for the major camphene products predicted; vapor pressures of products were calculated based on the Nannoolal method (Nannoolal et al., 2008). HOMs have much lower volatilities than the earlier terminal products such as NO3CAMP1, HO2CAMP1, and RO2CAMP1. HOMs formed by autoxidation steps in camphene radical chain reactions are mediated by the H-shift isomerization of $RO_2$-d and RO-d. Table 6 shows the SAPRC predicted fate of $RO_2$-a for all chamber runs; the fate of summed $RO_2$ is shown in Table S1, which includes $RO_2$-a~d and all the $RO_2$ radicals formed from other minor pathways. For the experiments without added $NO_x$ (WO1–6), once the initial peroxy radical $RO_2$-a was formed, a large fraction of $RO_2$-a (0.54-0.98) quickly reacted with $HO_2$ to form the terminal product HO2CAMP1, while only 2–3% of $RO_2$-a reacted through the NO pathway and led to the generation of HOMs. For the experiments with added $NO_x$ (W1–7), much higher $RO_2$-a + NO fractions (0.65–1.00) were predicted by SAPRC. The fates of summed $RO_2$ also suggested that not only $RO_2$-a, but also the other $RO_2$ radical intermediates would tend to favor further reactions through the NO reaction chain to form lower volatility products.

Based on the predicted fate of $RO_2$ in SAPRC simulations, the higher SOA mass yields in experiments with $NO_x$ were due to the formation of HOMs through autoxidation in the presence of $NO_x$. In general, faster $RO_2$ reaction with NO, $HO_2$ or other $RO_2$ limits HOM formation by autoxidation (Bianchi et al., 2019). In previous monoterpene SOA studies, HOM formation was often observed when $NO_x$ was absent or under lower $NO_x$ conditions (Pye et al., 2019; Schervish and Donahue, 2020; Zhao et al., 2018). For example, Zhao et al. (2018) demonstrated that autoxidation for some $RO_2$ is competitive with $RO_2$ + NO at ppb levels of NO for $O_3$-initiated α-pinene oxidation. They also reported that HOM formation decreased as the initial NO concentration increased from 0 ppb to 30 ppb. In the camphene experiments presented herein, the reverse trend was observed (see experiments WO4, W4 and W5 conducted with

~50 ppb camphene at different $NO_x$ levels). This was due to the key $RO_2$ species, $RO_2$-d, which was predicted to form only in the presence of $NO_x$ and had a fast enough autoxidation rate constant to effectively compete with bimolecular reactions.

While the decreasing SOA mass yields at high $[HC]_0$ and $M_o$ in experiments with and without added $NO_x$ (shown in Fig. 3) may not be statistically different within the uncertainty of the measurements, $RO_2$ chemistry was explored as an explanation for the apparent trends. For experiments with added $NO_x$, a shift in the $RO_2$ reaction pathways from NO to $HO_2$ can explain the decreasing SOA mass yields. The fraction of $RO_2$-a + NO decreased from 0.97 (W5) to 0.65 (W7) while the fraction of $RO_2$-a + $HO_2$ increased from 0.03 (W5) to 0.3 (W7). For the experiments without $NO_x$, the shift from $RO_2$ + $HO_2$ to self- and cross-reactions of $RO_2$ at high $[HC]_0$ and $M_o$ can explain the decreasing SOA mass yields. When $[HC]_0$ increased from 57 ppb to 223 ppb, the fractions of $RO_2$-a + $HO_2$ decreased from 0.89 (WO4) to 0.54 (WO6) and the fraction of $RO_2$-a + $RO_2$ increased by a factor of five, from 0.08 to 0.41. Moreover, this shift from bimolecular reactions with $HO_2$ to $RO_2$ as $[HC]_0$ increased also occurred in the context of the total $RO_2$ (Table S1). Generally, products that were predicted to form from one $RO_2$ reacting with another $RO_2$ in the absence of $NO_x$, had relatively higher volatility than those formed from that $RO_2$ reacting with $HO_2$; for example, RO2CAMP1 formed from $RO_2$-a + $RO_2$ was more volatile than HO2CAMP1 formed from $RO_2$-a + $HO_2$ (Table 5). The increasing fraction of self- and cross-reactions of $RO_2$ therefore is one likely explanation for the decreasing SOA mass yields at high $\Delta HC$ and $M_o$ in the experiments without $NO_x$.

The relatively low SOA mass yields in experiments W1 and W2 (0.36 and 0.33), also can be explained due to differences in product distribution. An underestimation of the SOA mass yields in these experiments due to the assumption of negligible wall loss is not sufficient to explain these relatively low yields. A comparison of the product distributions between W1, W2, W3 and W5 suggested similar yields of NO3CAMP1–5 and NOCAMP1–2, but major differences in yields of UNICAMP and HO2CAMP1–5 (Fig. S5). Experiments W3 and W5 were selected for comparison because of their closest total $RO_2$ fractional reaction distribution (approximately 90% $RO_2$ + NO and 10% $RO_2$ + $HO_2$) to W2 (98% $RO_2$ + NO and 2% $RO_2$ + $HO_2$) and W1 (96% $RO_2$ + NO and 4% $RO_2$ + $HO_2$) but higher SOA mass yield (0.64 and 0.6). W1 and W2 were predicted to have much smaller SOA mass yield than W3 and W5 in the low volatility products HO2CAMP1–5 (especially product HO2CAMP5, the lowest volatility among all listed products in Table 5, $\log_{10}C^* = -4.3$) and UNICAMP (the second lowest volatility shown in Table 5, $\log_{10}C^* = -3.9$), which can contribute to the lower SOA mass yield. Further analysis of W1 and W2 revealed a likely cause for the different yields of HO2CAMP1–5 and UNICAMP. W1 and W2 were predicted to have delayed peaks of [OH] (after 3–4 hours of irradiation) which likely was due to the high $NO_x$ concentrations (Fig. S1b). Correspondingly, the $[HO_2]$ was highly suppressed during the first 2 hours of irradiation. Under high $[NO_x]$, the $RO_2$-e + $HO_2$ pathway shown in Fig. 8 therefore could be suppressed, resulting in a lower yield of HO2CAMP5. This indicates a second "extreme $NO_x$" regime may exist at high $[NO_x]$ and significantly lower $[HC]_0/[NO_x]_0$.

## 5 GECKO-A simulations

### 5.1 SOA Mass and Yield

The comparison of gas- and particle-phase species between chamber experiments and GECKO-A model simulations
are shown in Fig. S1a and Fig. S1b. Without added $NO_x$, GECKO-A predicts much smaller camphene consumption
rates and no $O_3$ formation, while both the chamber data and SAPRC simulations suggest a final $O_3$ mixing ratio of
~10 ppb (Fig. S1a). This may be due to an underrepresentation of data and relevant pathways for low to no $NO_x$
conditions in the GECKO-A mechanism generation system, and the incomplete treatment of wall effects in the
application of the GECKO-A box model. The without added $NO_x$ simulations therefore are not further discussed. With
added $NO_x$, GECKO-A shows good agreement with the experimental data and SAPRC simulations in the context of
camphene consumption, $O_3$, and OH levels.

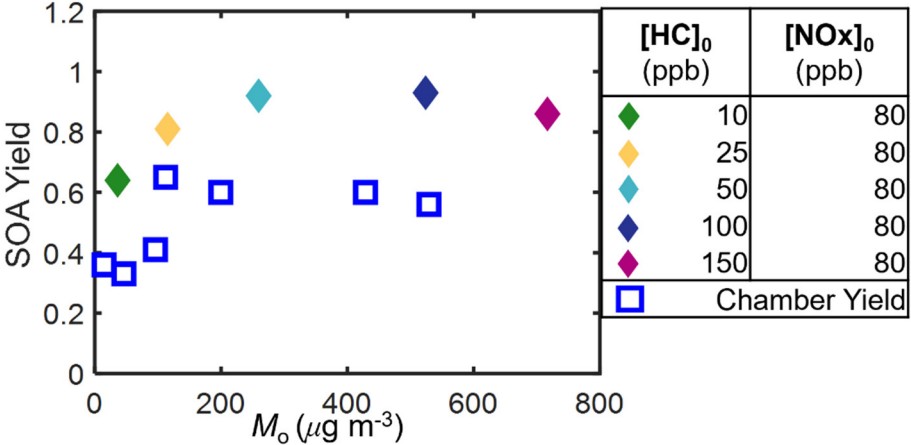

**Figure 9. Predicted SOA mass yields based on GECKO-A model simulations.**

Figure 9 shows the predicted SOA mass yields based on GECKO-A. For simulations with added $NO_x$, while the
model predicted higher SOA mass yields (0.64–0.93) than were observed (0.33–0.64), the trends in the SOA mass
yields were consistent between chamber observation and simulations. The simulated SOA mass yield increased with
SOA mass for SOA mass < 260μg m$^{-3}$, plateaued for SOA mass between 260 and 524 μg m$^{-3}$, and then decreased for
SOA mass > 524 μg m$^{-3}$.

Figure 9 shows the predicted SOA mass yields based on GECKO-A. For simulations with added $NO_x$, while the
The predicted O:C ratio and average carbon number (Fig. 10), defined as the mole-weighted averaged carbon
number for the main products (~95% by mass), were consistent with the plateauing/decreasing SOA yields at higher
$[HC]_0$ (Fig. 9). The average carbon number was calculated using equation (2):

$$\text{Average carbon number} = \frac{\sum_i \frac{nC_i \times M_{o,i}}{MW_i}}{\sum_i \frac{M_{o,i}}{MW_i}} \tag{2}$$

where $nC_i$, $M_{o,i}$, and $MW_i$ are the carbon number, mass, and molecular weight of species $i$, respectively. With added
$NO_x$, the average carbon number of both the gas and particle phases increased as $[HC]_0$ increased, while the O:C ratio
decreased. These trends indicate there is a significant fraction of higher volatility compounds formed that contribute
to SOA at higher $[HC]_0$ (or $M_o$), resulting in lower SOA mass yields. In addition, only at the highest two $[HC]_0$ were

non-negligible fractions of precursor predicted to react with $O_3$ and $NO_3$ (Fig. S7), suggesting a larger fraction of higher-volatility nitrogen-containing products. More detailed comparisons of GECKO-A simulations with chamber experiments are presented by Afreh et al. (2020) for camphene and McVay et al. (2016) for α-pinene.

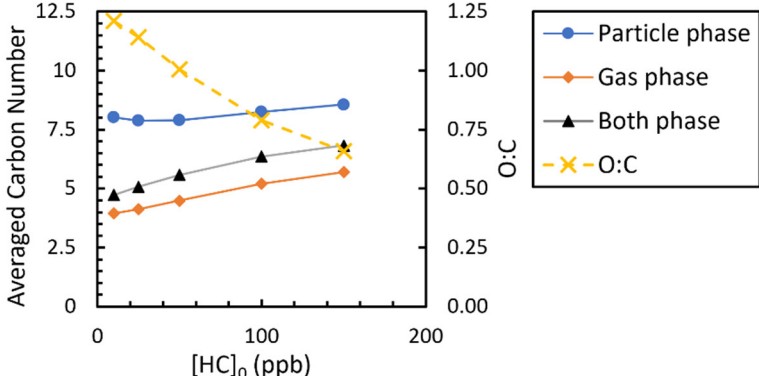


**Figure 10. GECKO-A predicted particle O:C and mole-weighted averaged carbon number of products with added $NO_x$.**

**5.2 Particle Density and O:C**

Figure 11a shows the GECKO-A predicted O:C ratio and measured O:C ratio as a function of $[HC]_0/[NO_x]_0$ for all experiments. A good agreement in O:C ratios was observed between the model predictions and chamber data. The

O:C ratio decreased from 1.21 to 0.39 as $[HC]_0/[NO_x]_0$ increased from 0.13 to 223, supporting that more highly oxygenated products were formed at lower $[HC]_0/[NO_x]_0$.

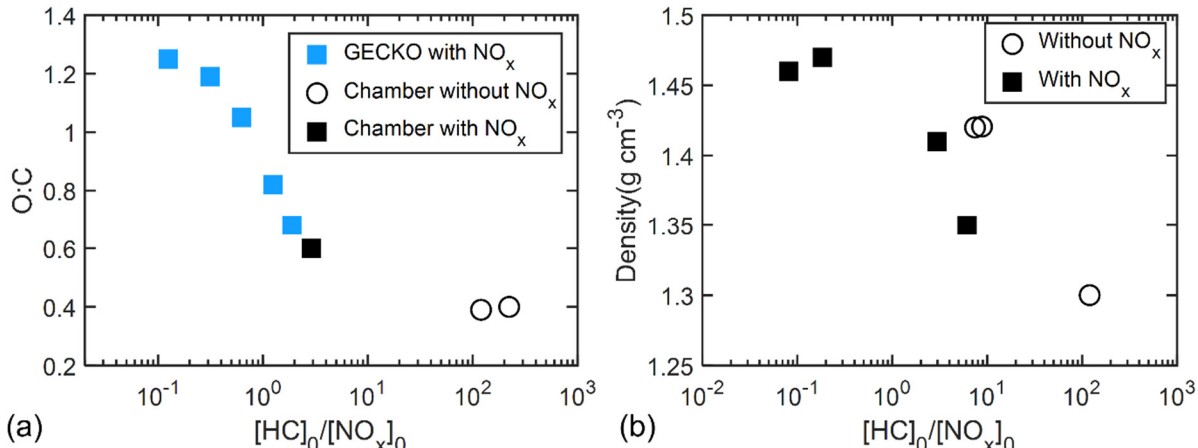

**Figure 11. (a), O:C ratio as a function of $[HC]_0/[NO_x]_0$ with AMS data and prediction by GECKO-A simulation. (b), Particle density (directly measured by APM–SMPS) shown as a function of $[HC]_0/[NO_x]_0$.**

A negative correlation was also observed between measured particle density and $[HC]_0/[NO_x]_0$. The final density of particles decreased from 1.47 g cm$^{-3}$ to 1.30g cm$^{-3}$ as $[HC]_0/[NO_x]_0$ increased from 0.08 to 120 (Fig.11b). The change in O:C ratio could account for the change in density. O:C and H:C have been used in semi-empirical SOA density parameterizations (Nakao et al., 2013; Kuwata et al., 2012), in which O:C plays a dominant role in determining organic particle density compared to H:C. Consistent with the semi-empirical formulations, the density of particles

formed from oxidation of camphene increased as O:C increased (from 0.39 to 1.21), while H:C varies over a smaller range (from 1.42 to 1.79). The change in density supports the proposed explanation that more oxygenated products were formed under lower $[HC]_0/[NO_x]_0$. The wide range in final density and the correlation with $[HC]_0/[NO_x]_0$ shown here has not been previously reported. The SOA mass of each experiment in this study was calculated with its own density of SOA, instead of applying an averaged density. A list of particle densities used in this study can be found in Table 2.

## 6 Conclusions

The first SOA mass yields from oxidation of camphene based on experiments performed in UCR environmental chamber with varying $[NO_x]_0$ are presented herein. Higher SOA mass yields were measured with added $NO_x$ (0.33–0.64) than without added $NO_x$ (0.08–0.26) at atmospherically relevant OH concentrations. SOA formation from the oxidation of camphene showed different $NO_x$ dependence than what has previously been reported for other monoterpenes (e.g., α-pinene, d-limonene) and n-alkanes (carbon≤ 10), in which higher SOA mass yields were measured when $[NO_x]$ was lower (Nøjgaard et al., 2006; Ng et al., 2007b). For camphene oxidation, higher $\Delta[HC]$ and lower $[HC]_0/[NO_x]_0$ (within 0.5–200) generally led to higher SOA mass yields. Similar $NO_x$ dependence has been observed for two sesquiterpenes (longifolene and aromadendrene) but was attributed to the production of nonvolatile organic nitrates with no detailed mechanistic analysis provided at that time (Ng et al., 2007b).

Although $[HC]_0/[NO_x]_0$ shows clear correlation with SOA mass yield, this quantity cannot completely represent the underlying $RO_2$ chemistry. The $RO_2$ chemistry and the competition between varying bimolecular $RO_2$ and unimolecular $RO_2$ reaction pathways, explored using SAPRC MechGen, can be used to explain the dependence of SOA mass yields on HC and $NO_x$. The $RO_2$ + NO pathway favored in experiments with added $NO_x$ formed HOMs with much lower volatilities than products formed in other pathways. In addition to the regular $NO_x$ regime introduced above ($[HC]_0/[NO_x]_0 > 0.5$), the results suggested an extreme $NO_x$ regime where high $[NO_x]$ may suppress SOA mass yield. High $NO_x$ levels may suppress $HO_2$ levels at the beginning of the experiments, causing a subsequent reduction in the yields of low volatility products such as UNICAMP and HO2CAMP5. This suggests that if the reactions happened in $NO_x$-rich environments with extremely high ratios of NO to $HO_2$ ($NO/HO_2$), the SOA mass yield from oxidation of camphene might be significantly suppressed. As demonstrated here, simulations with chemically detailed box models such as SAPRC are useful for identifying SOA formation regimes.

Overall, SOA formation from oxidation of camphene may be larger in polluted environments (e.g., urban environments) than $NO_x$-free environments. This reveals a possible underestimation of SOA formed from oxidation of camphene and potentially other VOCs that are assumed to have lower SOA mass yields at higher $NO_x$ levels. Further chamber and modeling studies of other understudied VOCs will be important for identifying other systems in which moderate $NO_x$ levels can promote HOM formation.

## Data Availability

The experimental and modeling data is available upon request from the corresponding authors.


**Supplement**

The supplement related to this article is available and can be download from ACP assigned link.

**Author Contributions**

QL and JJ contributed equally to the study and share the first authorship. QL performed chamber experiments, data analysis and led the first draft of the manuscript. JJ derived and implemented the camphene mechanism in SAPRC, conducted SAPRC model simulations and led discussions on the chemistry of camphene SOA formation. IA carried out GECKO-A model simulations. QL and JJ interpreted the results and wrote the manuscript with IA. QL, JJ, KB and DC finalized the final manuscript. All the listed authors contributed to the revisions of the manuscript. The project 520 was supervised by KB and DC.

**Competing Interests**

The authors declare that they have no conflict of interest.

**Acknowledgements**

This study received support from the National Science Foundation grant AGS-1753364. The authors acknowledge Dr. William Carter for helping with the discussions in the MechGen estimation methods.

This publication was developed under Assistance Agreement No. 84000701 awarded by the U.S. Environmental Protection Agency to University of California Riverside. It has not been formally reviewed by EPA. The views 530 expressed in this document are solely those of the authors and do not necessarily reflect those of the Agency. EPA does not endorse any products or commercial services mentioned in this publication.

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
