# Peer review of "Secondary Organic Aerosol Formation from Camphene Oxidation: Measurements and Modeling"

_Atmospheric Chemistry and Physics, 2021_

## Author Comment (AC1)

**Response to Reviews**

**Secondary Organic Aerosol Formation from Camphene Oxidation: Measurements and Modeling**

Qi Li[1,2], Jia Jiang[1,2], Isaac Kwadjo Afreh [1,2], Kelley C. Barsanti[1,2], David R. Cocker III[1,2]

[1]Department of Chemical and Environmental Engineering, University of California-Riverside, Riverside, California 92521, United States
[2]The Bourns College of Engineering, Center for Environmental Research and Technology, University of California-Riverside, Riverside, California 92507, United States

*Correspondence to*: Kelley C. Barsanti (kbarsanti@engr.ucr.edu) and David R. Cocker III (dcocker@engr.ucr.edu)

We thank the reviewers for their thoughtful questions and comments, which have served to improve both the clarity and quality of the manuscript. Our responses to the questions and comments are included below and highlighted using blue text.  We refer to the **tracked changes manuscript** as **"TM"** and **tracked changes supporting information** as **"TS".**

**Reviewer 1**

**Summary:**

This manuscript describes a series of experiments and simulations designed to investigate the variables upon which SOA formation from the OH-initiated oxidation of camphene depend in the ambient atmosphere. In particular, the authors investigate the NOx-dependence of SOA yields, and find that increasing NOx counterintuitively increases
SOA yields. Mechanistic modeling enables the authors to pinpoint the source of this effect via elevated yields of highly oxidized molecules produced from a series of b-scissions and oxygen additions following the reaction of the initial camphene + OH + O2 peroxy radical with NO. Furthermore, under "extreme" high-NOx conditions, SOA yields decrease, which the authors again hypothesize is due to the nuances of the camphene oxidation mechanism, whereby the exclusive reaction of peroxy radicals with NO can eventually form more-volatile products. These effects are at
times complicated by the other changing variables between experiments, including the initial hydrocarbon loading and the SOA mass formed, both of which can change vapor-wall effects, but the authors are able to make a compelling case for the mechanistic reasoning behind NOx-dependent SOA yields.

Now that I read the rest of my comments below, it sounds like a lot of complaints, but I really think this an an excellent synthesis of experiments and modeling and an important step in our understanding of how RO2 fates influence
important outcomes like SOA formation from VOCs in the atmosphere. It's great to see such a comprehensive study and with such complementary modeling and experimental parts that both bring a lot to the table -- in particular, the way the mechanistic modeling is able to explain the complex NOx dependence of the SOA yield. More comprehensive consideration of both sources of uncertainty and vapor wall effects could make this a still stronger paper, but it's great already!

Thank you. We appreciate your comments and have addressed individual comments below.

**Major comments:**

1. The lack of consideration of vapor wall effects is puzzling. The authors cite two studies that saw little difference between seeded and unseeded experiments, but any effects would still be highly dependent on the initial hydrocarbon loading (how does that compare to the other studies?) and the precise details of the oxidation mechanisms leading to SOA formation. Because the conclusion of this paper is precisely that camphene's SOA-formation mechanism is *different* from many other VOCs, with a predominantly positive NOx dependence, it's not clear that we should be able to extrapolate from other VOCs' vapor-wall effects (especially if other VOC's make a lot of SOA from low-volatility products like dimers, while camphene's are intermediate-volatility compounds). One could just as well cite plenty of studies that do see a strong effect of initial seed surface area on measured SOA yields, which are demonstrably due to the competition between vapor-wall and vapor-particle partitioning (e.g. Zhang et al., 2014 & 2015; Schwantes et al, 2019). It is not clear from the sources cited here that the same effects aren't at play in the UC Riverside chamber.

The positive dependence of SOA yield on dHC (L 252) and M(0) (L 225), especially at low values of dHC and M(0), could easily be explained by wall effects, whereby lower initial SOA formation leads to higher losses of compounds that would otherwise form SOA to the walls instead of to (newly formed) particles in experiments with lower [HC]0 and M(0). This could also compound the effects of the low SOA yields at low NOx -- if the yields are slightly lower at low NOx, the reduced initial particle formation leads to greater losses of SOA precursors to the walls rather than to particles, which thus leads to an even lower measured SOA yield. Vapor-wall effects will therefore have a tendency to exaggerate any observed differences in SOA yields. The authors seem to admit this might be a problem on L 380-381, describing model-measurement discrepancies.

I don't mean to suggest that the authors need to start from scratch or perform a whole new set of experiments to see how [HC]0 or the introduction of seed particles might change observed yields, although either would be extremely interesting. But some discussion of the effects that vapor wall losses could play here is certainly merited, along with how it would change the conclusions drawn from the observations.

Thanks to the reviewer for the points and suggestions. We acknowledge the competition between vapor-wall and vapor-particle partitioning that has been observed in chamber-based SOA experiments, and agree that, as a consequence, vapor wall loss can contribute to an underestimation of SOA yield, particularly at low mass loadings. However, in the UCR chamber, we have not observed measurable differences in SOA formation between seeded and non-seeded experiments of other compounds, including α-pinene, m-xylene, and benzyl alcohol (a typical LVP-VOC, unpublished) in the UCR chamber, thus covering a range of precursor and product volatilities. Based on these prior experiments in the same chamber, it is expected that vapor-particle partitioning is the dominant process in camphene experiments under varying $[HC]_0$. We have added acknowledgments for other chamber works on vapor wall loss (e.g. Zhang et al., 2014 & 2015; Schwantes et al, 2019) and have added further details on our wall loss characterizations and assumptions.

**Lines 127- 134 in TM** (tracked changes manuscript) **now read**: **"Vapor wall loss of organics has been reported in multiple chambers (e.g., Zhang et al., 2015, 2014; Schwantes et al., 2019). In the UCR chamber, vapor wall loss has been investigated in SOA experiments using various precursor compounds (including α-pinene and *m*-xylene) under seed and no seed conditions (Clark et al., 2016; L. Li et al., 2015); no measurable differences in SOA formation have been observed in any of these experiments indicating negligible vapor wall losses. In this work, stability tests on camphene also resulted in negligible vapor wall loss of the parent compound. Thus, the assumption of negligible vapor wall loss was maintained for these experiments. It is noted that this assumption does not affect the major conclusions regarding the role of gas-phase chemistry on SOA formation."**

Regarding vapor wall loss at low $[HC]_0$ ($M_o$), we agree with the reviewer that significant wall losses would result in an underestimation of SOA yield particularly at low $[HC]_0$ ($M_o$). For the results presented here, experiments W1, W2,

WO1, and WO2 (see figure below) have the lowest SOA mass yields. An increase in the SOA yields of these experiments, would better align the SOA yields with the RO₂/HC ratios and would not affect the discussion or conclusions regarding the relationship between RO₂ chemistry and SOA mass yields. However, since we did not explicitly conduct wall loss experiments as part of these studies, we acknowledge the reviewer's comments as follows (**lines 331-334 TM**):

**"Though vapor wall loss has been found to be negligible in previous UCR chamber experiments, such experiments were typically conducted at higher [HC]₀. Thus, it is acknowledged that vapor wall loss could affect the measured SOA yields, particularly for experiments W1-2 and WO1-2 with low [HC]₀ (or $M_0$). A vapor wall loss correction for those experiments would increase the measured SOA, but would not affect the following discussion or conclusions regarding the role of RO₂ chemistry."**

[Figure]

**Figure 7. Relationship between total [RO₂]/[HC]₀, [HC]₀/[NOₓ]₀, and SOA mass yields.**

References:

Schwantes, R. H., Charan, S. M., Bates, K. H., Huang, Y., Nguyen, T. B., Mai, H., Kong, W., Flagan, R. C., and Seinfeld, J. H.: Low-volatility compounds contribute significantly to isoprene secondary organic aerosol (SOA)
under high-NOx conditions, Atmos. Chem. Phys., 19, 7255–7278, https://doi.org/10.5194/ACP-19-7255-2019, 2019.

Zhang, X., Cappa, C. D., Jathar, S. H., McVay, R. C., Ensberg, J. J., Kleeman, M. J., and Seinfeld, J. H.: Influence of vapor wall loss in laboratory chambers on yields of secondary organic aerosol, Proc. Natl. Acad. Sci., 111, 5802–5807, https://doi.org/10.1073/PNAS.1404727111, 2014.

Zhang, X., Schwantes, R. H., McVay, R. C., Lignell, H., Coggon, M. M., Flagan, R. C., and Seinfeld, J. H.: Vapor wall deposition in Teflon chambers, Atmos. Chem. Phys., 15, 4197–4214, https://doi.org/10.5194/ACP-15-4197-2015, 2015.

2. The concept of the "extreme NOx regime" is introduced slowly and in such a way that some of the earlier claims in the paper don't seem supported by the data, or at least aren't clear until much later on. The "extreme NOx regime" is
mentioned briefly at L 257 but not explained until later, so many of the earlier statements -- that SOA yield is high when there's added NOx, or that it depends on M(0), for example -- at first seem misleading when the accompanying figures show that above a certain point, added NOx seems to decrease yields. The payoff only comes around page 17 when the chemical reasoning behind the decreased SOA yields in W1 and W2 is explained. I'm not suggesting a complete restructuring of the paper, but I think it could be improved if this chemical explanation were more concretely hinted at earlier, and if the reduced SOA formation at "extreme" NOx were mentioned in the abstract as well.

As an example, at L 221-227, it sounds like the lower SOA yield in W7 relative to W6 and WO6 relative to WO5 will be a dependence on M(0). I understand that it's tough to put everything in an order that explains it all clearly at once, but Fig 2 is particularly misleading because it and the associated discussion makes it sound like this is going to be a dependence on SOA mass, but only much later do you explain it's actually a dependence on RO2 fate, where the high RO2+RO2 chemistry in WO6 and "extreme" NOx chemistry in W7 decrease yields. (As a side note, given the few points on this graph and the fact that WO5 and WO6 have very similar M(0), it almost doesn't seem like you can say there's a "trend" toward lower SOA yields at highest M(0) levels). It would be helpful to briefly mention here what the actual dependences are, even if you'll wait until later sections to explain them more fully.

We appreciate these points and suggestions. We decided to structure the manuscript to first address well understood factors that influence SOA yield, including reacted HC concentrations and SOA mass formation, and followed by a discussion of $RO_2$ chemistry. In response to this comment, an acknowledgment of the extreme $NO_x$ condition has been added to the abstract, **lines 30-31 in TM: "Further analysis reveals the existence of an extreme $NO_x$ regime, where the SOA yield can be suppressed due to high NO/HO₂ ratios".** In addition, a corresponding edit has been made to the suggested section (**lines 273-274 in TM**) to introduce the $RO_2$ chemistry effects earlier in the manuscript and build a better connection with the detailed discussion of $RO_2$ chemistry: **"These unexpected trends in SOA mass yields were further investigated and largely explained by the RO₂ fate based on box model simulations (see Sect. 4 & 5)."**

3. Uncertainties and replicability -- on the topic of Figure 2, it would be much easier to assess whether W7 and WO6 represent a decreasing trend at high M(0) if we had some estimate of uncertainty on either axis, ideally in the form of error bars. Overall, this paper could benefit from more discussion of the potential places where experimental or modeling uncertainties may confound the interpretation of results. On the experimental side of things, how replicable are wall-loss experiments, and therefore how much error is introduced by the wall-loss corrections, which would presumably carry through to SOA yield? On the model side, how well-constrained are the rates of the RO2 reactions that allow you to estimate the branching fractions in Figure 5, and how well constrained are the product yields in Figure 7? If possible, this could be described along with the instrument and model descriptions in the methods section, and uncertainty ranges could be added onto numbers reported in tables (e.g. Table 2) and/or error bars added to figures.

Regarding the reproducibility of the measured SOA yields, we have previously characterized the uncertainties of this chamber system by running a set of repeated experiments; 10 repeated m-xylene oxidation experiments showed an SOA yield uncertainty of $< 6.65\%$ (Li et al., 2016). We added the following statement to **lines 144-146 in TM: "Based on a prior characterization of this UCR chamber system (Li et al., 2016), the experimental uncertainty in measured SOA yield is < 6.65%".**

The MechGen-derived $RO_2$ rate constants and mechanisms are based on a wealth of reported experimental data and estimations methods in which experimental data are not available. The references and estimations methods are described briefly on the SAPRC website: https://intra.engr.ucr.edu/~carter/SAPRC/18/. For the $RO_2$ radicals that are represented explicitly (RO2-a ~ RO2-e) in the mechanism, their rate constants were calculated individually based on their structures. For the other $RO_2$ radicals counted in the total $RO_2$, their rate constants were derived as an alkyl $RO_2$ with 5 carbons which may lead to an underestimation of reaction rates for bigger molecules like camphene. The following statement was added to **lines 240-247 in TM: "The quantity $\Delta([O_3] - [NO])$ has been used to evaluate the rate of NO oxidation by RO₂ for VOC-NOₓ systems in SAPRC mechanism development (Carter and**

**Lurmann, 1990; Carter, 1999; Carter, 2009; Carter, 2020). Figure S2 shows the comparison of the $\Delta([O_3] - [NO])$ values between chamber measurements and SAPRC simulations for experiments with added $NO_x$. The SAPRC box model captures the rates of $RO_2$+NO well, and supports the use of SAPRC model to interpret chamber observations especially in the presence of $NO_x$. Unfortunately, it is hard to quantify how well constrained the other $RO_2$ reaction rates and product yields are without corresponding measurements, which are not available.**

**In this case, the SAPRC model was largely used to probe the mechanism (diagnostic) and not to predict yields (prognostic)."** Figure S2 was added in TS:

[Figure]

**Figure S2. Comparison of the chamber data (circles) and SAPRC model simulation results (lines) for camphene photo-oxidation experiments with added $NO_x$.**

References:

Li, L., Tang, P., Nakao, S., and Cocker III, D. R.: Impact of molecular structure on secondary organic aerosol formation from aromatic hydrocarbon photooxidation under low-$NO_x$ conditions, Atmos. Chem. Phys., 16, 10793–10808, https://doi.org/10.5194/acp-16-10793-2016, 2016.

Carter, W. P. L. and Lurmann, F.W., 1990. Evaluation of the RADM gas-phase chemical mechanism. US
Environmental Protection Agency, Atmospheric Research and Exposure Assessment Laboratory, Office of Research and Development.

Carter, W. P. L. Documentation of the SAPRC-99 Chemical Mechanism for VOC Reactivity Assessment, 1999; p 329.

Carter, W. P. L. Development of the SAPRC-07 Chemical Mechanism and Updated Ozone Reactivity Scales; Final Report to the California Air Resources Board Contract No. 03-318, March 2009.

Carter, W. P. L. Documentation of the SAPRC-18 Mechanism; Report to California Air Resources Board Contract No. 11-761, May 2020.

A corollary to this is that sometimes the places with the most uncertainty and model-measurement disagreement are the most interesting to dig into, because they have the potential to show what is lacking in our current understanding of the chemistry in question. To that end, I think the statements about model-measurement disagreement on L 377 & 387 deserve more explanation. First, what could be causing the big differences at low NOx between GECKO simulations and observations? OH recycling, or higher background NOx? And second, why might the modeled absolute SOA yields with added NOx be overestimated by up to a factor of 2? How much could this be due to wall losses, uncertain VBS parameters, or the mechanism itself? I know these model-measurement differences may seem too big to tackle here and like they're beyond the scope of the paper, but even just some speculation thrown in here could be useful to guide the reader's thinking!

GECKO-A, and the underlying SARs, largely have been developed and tested for moderate to high $NO_x$ levels. The ability of the GECKO-A mechanism generation system to represent low $NO_x$ conditions is largely untested and thus, as noted in the manuscript, these simulations are presented but not further discussed. In the absence of prior evaluations and appropriate gas-phase measurements for these studies, it would be too speculative to try to explain the differences at low $NO_x$ conditions. We do note that the addition of a constant low level of $NO_x$ (to represent the $NO_x$ off gassing in the chamber) did not significantly change the GECKO-A model predictions under low $NO_x$ conditions.

Regarding the predictions under added $NO_x$ conditions, some general differences between the GECKO-A model simulations and chamber experiments include: initial conditions, no representation of H-shift reactions in GECKO-A (not available in the current version), no consideration of wall losses in GECKO-A, and uncertainties in vapor pressure predictions in GECKO-A. In addition, GEKCO-A assumes equilibrium gas/particle partitioning and does not include condensed phase reactions. Clearly some of these differences could decrease the differences and others could increase the differences. We note there are some differences in the branching ratio profiles presented in the SI (Fig. S1b). In response to this comment, we have included the following sentence, with reference to prior more detailed comparisons between GECKO-A model predictions and chamber measurements **(lines 474-475 TM): "More detailed comparisons of GECKO-A simulations with chamber experiments are presented by Afreh et al. (2020) for camphene and McVay et al. (2016) for α-pinene."**

Reference:

Afreh, I. K., Aumont, B., Camredon, M., and Barsanti, K. C.: Using GECKO-A to derive mechanistic understanding of SOA formation from the ubiquitous but understudied camphene, Atmos. Chem. Phys. Discuss., https://doi.org/10.5194/acp-2020-829, 2020.

McVay, R. C., Zhang, X., Aumont, B., Valorso, R., Camredon, M., La, Y. S., Wennberg, P. O., and Seinfeld, J. H.: SOA formation from the photooxidation of α-pinene: Systematic exploration of the simulation of chamber data, Atmos. Chem. Phys., 16, 2785–2802, https://doi.org/10.5194/acp-16-2785-2016, 2016.

**Other comments:**

L 96: What is "2mil"?

MIL is a manufacturing measurement unit. 1 MIL= 1/1000 inch= 0.0254 mm. We have added "2 MIL (0.0508 mm)".

L 183: What does "final peak particle diameter" mean? Is it the highest-diameter particle measured or the median/mean particle diameter at some "final" time?

The "final" time is equivalent to the time at the end of experiment and is determined by the point at which the chamber collapses and cannot keep the positive pressure difference (~0.015 in $H_2O$ = 3.73 Pa) to the ambient pressure. The peak particle diameter refers to the diameter of particles shown at the peak of the size distribution plot at the end of the experiment. Footnotes have been added to **Table 2 in TM to clarify: "Peak $d_p$ refers to the diameter of particles at the peak of the size distribution plot at the end of the experiment. The uncertainty of peak $d_p$ values is less than 5%.**"

L 184: Here and throughout, it would be helpful to be more specific with the definition of "SOA yield". Is it the mass yield or a molar yield assuming a chemical identity for the SOA-phase compound(s)? Is it the yield measured at its maximum, the end of the experiment, or a specified photochemical aging time? Even if you define it once somewhere in the paper, to avoid confusion it's nice to consistently refer to it as specifically as possible (e.g. as "peak SOA mass yield") wherever it's subsequently brought up.

With the exception of the SOA yields shown in Figure 4, all SOA yields discussed in the paper are mass based yields and were calculated at the end of the experiment (~ 6 hours). We have replaced all appearances of "SOA yield" with

"SOA mass yield" in the figure titles and text. The following statement was added to **line 218 in TM: "Except for Fig. 4, in which SOA mass yields are shown as a function of photochemical age, all SOA mass yields refer to the mass at the end of the experiments (~6 hours).**"

Figure 1: Agreement between measured and modeled values would be much easier to see if c and e were plotted together; same with d and f.

Thank you for the suggestion. Figure 1 (now Fig. 2) was updated in TM.

[Figure]

**Figure 2. SAPRC predicted β values: (a) without added NOₓ, and (b) with added NOₓ. Measured (circles) and predicted (lines) camphene consumption as a function of irradiation time: (c) without added NOₓ, and (d) with added NOₓ. The hollow makers used in (c) and (d) are equivalent to dashed lines defined in the legends.**

L 243-244: The claim that the SOA yield curves "already plateau or nearly plateau by the end of experiments" doesn't seem to be supported by Figure 3, where all the high-NOx experiment yield curves are flat or even decreasing (how can that be explained, by the way?!) by the end of the experiment, whereas every single low-NOx experiment yield curve still has a positive slope. Based on the change in slopes, how long might it take for the low-NOx experiments to plateau, and how much higher could their yields rise? Without knowing that, it seems an apples-to-apples comparison might cut off all the experiments at the same approximate photochemical aging time and see how they differ -- but cutting off some of the high-NOx experiments at ~15 h photochemical age to better compare to the low-NOx experiments' maxima could cause a considerable change in reported yields, even bringing W1 to a "final" SOA yield lower than that of some of the low-NOx experiments. How much would extrapolating the low-NOx yield curves to high aging times where they plateau, or conversely cutting off the high-NOx yield curves at much lower aging times, change the analysis in this paper?

We agree with the reviewer that the trends with photochemical age are difficult to compare in the figures as presented. In response to these comments, Figure 4 has been updated to include all experimental data in one figure. When plotted on the same scale, it is clearer that most of the yield curves have plateaued or nearly plateaued. The "except for some of the experiments without added NO$_x$" has been deleted. The slightly decreasing trend of W6 in Figure 3 (now Figure 4) was due to timeline drift and has been corrected. We have accepted the reviewer's suggestion to compare SOA yields over the same aging time. The additional figure has been to the TS (Fig. S8). Together these figures make it clearer that the SOA yields were higher with added NO$_x$ than without and that even for the without added NO$_x$ experiments that may not have completely plateaued, they are not close to the yields of the with added NO$_x$ experiments. The text now reads (lines 285-290 TM):

**"The SOA mass yields are dependent on OH levels and thus photochemical aging time. The yield curves plateaued or nearly plateaued for most experiments by the end of the experiment. Higher [HC]$_0$ generally led to steeper increases in SOA mass yield as a function of aging time. Experiments with added NO$_x$ generally had longer photochemical aging times than experiments without added NO$_x$; even at the same aging time (Fig. S8), the SOA yields were higher in the with added NO$_x$ experiments."**

[Figure]

L 270-271: It's unclear to me what the "accumulated total [RO2]" is measuring or is useful for. Does this count each b-scission-plus-O2 step as an independent production of RO2 toward the cumulative total? In this case, it's kind of conflating the fraction of hydrocarbon reacted with the number of b-scission reactions per camphene+OH reaction, right? Since it's not further discussed (unless I'm missing something) I'm not sure why it's brought up here.

Thanks to the reviewer for pointing this out. The total RO$_2$ in the previously submitted version was calculated based on a model counter species, which was used to represent the sum of concentrations of all RO$_2$ species which included the RO$_2$s formed through the β-scission-plus-O$_2$ steps. Besides that, this counter also included the other RO$_2$s that could be formed from camphene + NO$_3$/O$_3$ and the oxidation reactions of the products formed by camphene. The previous calculation of total $RO_2$ overcounted the unimolecular reactions and should not be used to compare with bimolecular reactions. The purpose of showing total $RO_2$ fate is using it as an indicator of chemical conditions, or more specifically, the overall ratios of $NO:HO_2:RO_2$ during the experiment. Thus, the current total $RO_2$ was recalculated and updated as the summation of $RO_2$ undergoing bimolecular reactions. The updates do not affect the original conclusions but deliver the information more clearly.

Changes: Table S1 in TS was modified by deleting the "Uni" column and merging "$RCO_3$" and "$RO_2$" columns. The title of Table S1 was modified as: "Weighted fractions of total peroxy radical bimolecular reactions of each type, calculated based on SAPRC simulations." Footnotes were added: "[a] "$RO_2$s" refers to the sum of reactions of $RO_2$ with $RO_2$ and with $RCO_3$." The definition of total $RO_2$ was added to **lines 316-317 in TM: "total RO₂ (calculated as the summation of RO₂ that undergo bimolecular reactions)".** Figure 6 and Figure 7 with corresponding discussion in section 3.3 were updated accordingly to reflect the changes. All the other discussions that mention total $RO_2$ have been revised using the current definition of total $RO_2$.

L 290: Needs a comma, not a semicolor

Corrected, thank you.

L 300: Since there's no aromaticity, this compound can't be described as phenolic. It's an alcohol, though.

Corrected, thank you.

L 304: "Peroxy", not "proxy"

Corrected, thank you.

Figure 7: The compound produced in the +NO (0.806)/+NO3/+RO2 (0.5)/+RCO3 pathway from RO2-e should be an alkoxy radical; the way it's drawn, it looks like a stable compound.

Thank you for pointing out the mistake. It is now corrected in Fig. 7 (now Fig. 8) (TM) and Figure S4 (TS).

Also, there is some indication that RO2 + HO2 reactions of large and/or functionalized peroxy radicals can produce reasonably high yields of alkoxy + O2 + OH rather than the radical-terminating hydroperoxide ROOH, although it seems this mechanism assumes 100% ROOH formation (see, e.g., Praske et al. 2015, Kurten et al. 2017). How would this pathway change the model interpretation?

Though we are aware of the recent works on $RO_2$ + $HO_2$ reactions, the implications haven't been assessed yet in MechGen. The current version of MechGen do not predict the alkoxy + $O_2$ + OH pathway except for acyl peroxy radicals. The camphene $RO_2$s (except for RO2-e) are considered to be alkyl peroxy radicals that will only form ROOH with 100% yield. While modifying MechGen is beyond the scope of this work, we can assume that since we injected approximately 1 ppm of $H_2O_2$ as the OH source, any OH radicals formed from this unrepresented pathway would have a negligible influence on the gas-phase chemical conditions ($NO:HO_2:RO_2$). However, there might be an overestimation of ROOH should this pathway be important under the modeled conditions.

L 435: This sentence is confusing and appears to have a grammar issue. Maybe replace the "but" with ", it"?

Corrected, thank you.

L 438: "experiment" should either be plural or replaced with "the experiment"

Corrected, thank you.

L 438: How did the RO2 + NO pathway lead to the highest RO2 production? Is this because it had higher OH and therefore more camphene reacted, or is this referring to the "accumulated total [RO2]/[HC]0" discussed above (see comment on L 270-271)

Yes, it was referring to the accumulated total $[RO_2]_0/[HC]_0$. To avoid confusion, we deleted it from the conclusion.

L 443: Why is the ratio in parentheses presented in the opposite order to the way it's described here?

Corrected, thank you.

L 462-463: Is "IS" supposed to be "IA"?

Corrected, thank you.

References:

Kurtén, T.; Møller, K. H.; Nguyen, T. B.; Schwantes, R. H.; Misztal, P. K.; Su, L.; Wennberg, P. O.; Fry, J. L.; Kjærgaard, H. G., Alkoxy Radical Bond Scissions Explain the Anomalously Low Secondary Organic Aerosol and Organonitrate Yields from α-Pinene + NO3. J. Phys. Chem. Lett. 2017, 8, 13, 2826–2834.

Praske, E., Crounse, J. D., Bates, K. H., Kurtén, T., Kjaergaard, H. G., Wennberg, P. O. Atmospheric fate of methyl
vinyl ketone: peroxy radical reactions with NO and HO2. J. Phys. Chem. A, 119 (19), 4562-4572. DOI: 10.1021/jp5107058, 2015.

Schwantes, R. H., Charan, S. M., Bates, K. H., Huang, Y., Nguyen, T. B., Mai, H., Kong, W., Flagan, R. C., and Seinfeld, J. H.: Low-volatility compounds contribute significantly to isoprene secondary organic aerosol (SOA) under high-NOx conditions, Atmos. Chem. Phys., 19, 7255–7278, DOI: 10.5194/acp-19-7255-2019, 2019.

Zhang, X., Cappa, C. D., Jathar, S. H., McVay, R. C., Ensberg, J. J., Kleeman, M. J., and Seinfeld, J. H.: Influence of vapor wall loss in laboratory chambers on yields of secondary organic aerosol, Proc. Nat'l. Acad. Sci., 111, 5802–5807, 2014.

Zhang, X., Schwantes, R. H., McVay, R. C., Lignell, H., Coggon, M. M., Flagan, R. C., and Seinfeld, J. H.: Vapor wall deposition in Teflon chambers, Atmos. Chem. Phys., 15, 4197–4214, DOI: 10.5194/acp-15-4197-2015, 2015.

**Reviewer 2**

**Summary:**

In this work, the authors studied oxidation of camphene and the resulting secondary organic aerosol (SOA) formation. Most studies have shown that monoterpene SOA yields decrease with increasing NOx, but this study shows the opposite for camphene. To understand this trend the authors combined chamber experiment results with detailed gas- phase (SAPRC) and aerosol formation (GECKO-A). They showed that NO increases the formation of radical intermediates that can isomerize rapidly to form highly oxygenated molecules (HOMs) which have very low volatilities. This study is beautifully done and provides an elegant explanation to a complex phenomenon. I am particularly impressed with how the authors integrated modeling with experimental results and provide a fundamental understanding of this system. I highly recommend publication, after addressing the following minor comments:

Thank you for the kind words. Individual comments are addressed below.

The only overall question that I have is how this can be generalized to other systems. What is unique about camphene that NO actually increases the formation of HOMs? We tend to think that NO and HO2 promotes termination reactions, but in this case NO turns the radicals into an "isomerizable" form. Is this unique to camphene, or should we start looking for these pathways in other systems? Could this happen to, for example, sesquiterpenes, which may be an alternate explanation to the higher yields under higher NOx?

We are asking ourselves the exact same question and plan to address it in future work. We do not think that camphene is entirely unique, and it is likely that this chemistry occurs in other molecules with similar structures. In the context of sesquiterpenes specifically, unfortunately because MechGen does not support parallel computing as it is currently configured, it is not capable of treating large molecules (including sesquiterpene).

**Other comments:**

Line 36: "14% of the total reactive VOC flux", is that 14% of the reactivity, or 14% of the mass emitted?

It was based on mass, Tg C. The associated sentence is updated to include the unit.

Section 1 Introduction: the literature review is concise and relevant. As a reader who does not think about camphene regularly, I would find some background information about camphene to be useful. For example, what is its OH rate constant, and how does its reactivity compare to other monoterpenes? Also I do not see its molecular structure until Figure 7. I personally like to visualize the molecule (its bicyclic structure, 1 C=C double bond) while reading the introduction so there is a better context.

Thanks for the suggestions. A new Figure 1 was added to show the chemical structure and reaction rate constants of camphene:

Figure 1. Camphene chemical structure and reaction rate constants (unit: cm³ molecule⁻¹ s⁻¹) with major atmospheric oxidants.

Lines 70-84: given the results of this study showing the importance of HOM, it might be useful to mention the recent knowledge about RO2 autoxidation as an important pathway for RO2 radicals too (e.g. Crounse et al., J Phys Chem Lett, 2013 and many others).

Thank you for the suggestion. We added a paragraph to **lines 90-98 in TM (**tracked changes manuscript) to talk about the importance of RO$_2$ autoxidation and HOMs**: "The atmospheric gas-phase autoxidation of RO$_2$ has been identified as another key pathway of SOA formation (Crounse et al., 2013; Jokinen 2014; Ehn et al., 2017; Bianchi et al., 2019). The RO$_2$ radical undergoes intramolecular H-atom abstraction reactions to form a hydroperoxide functionality and an alkyl radical (RO), to which a new RO$_2$ will be formed by adding O$_2$. The autoxidation process can repeat several times until terminated by other pathways and will form low-volatility compounds known as highly oxygenated organic molecules (HOMs) (Bianchi et al., 2019). Recent theoretical and experimental studies have been conducted to understand HOM formation from monoterpenes such as α-pinene and β-pinene (Zhang et al., 2017; Quéléver et al., 2019; Xavier et al., 2019; Pullinen et al., 2020; Ye et al., 2020), but the potential importance and mechanisms of HOM formation from camphene have not been well investigated."**

References:

Crounse, J. D., Nielsen, L. B., Jørgensen, S., Kjaergaard, H. G., and Wennberg, P. O.: Autoxidation of Organic Compounds in the Atmosphere, J. Phys. Chem. Lett., 4, 3513–3520, https://doi.org/10.1021/JZ4019207, 2013.Ehn, M., Berndt, T., Wildt, J., and Mentel, T.: Highly Oxygenated Molecules from Atmospheric Autoxidation of Hydrocarbons: A Prominent Challenge for Chemical Kinetics Studies, Int. J. Chem. Kinet., 49, 821–831, https://doi.org/10.1002/KIN.21130, 2017.

Pullinen, I., Schmitt, S., Kang, S., Sarrafzadeh, M., Schlag, P., Andres, S., Kleist, E., Mentel, T. F., Rohrer, F., Springer, M., Tillmann, R., Wildt, J., Wu, C., Zhao, D., Wahner, A., and Kiendler-Scharr, A.: Impact of NOxon secondary organic aerosol (SOA) formation from α-pinene and β-pinene photooxidation: The role of highly oxygenated organic nitrates, Atmos. Chem. Phys., 20, 10125–10147, https://doi.org/10.5194/ACP-20-10125-2020, 2020.

Quéléver, L. L. J., Kristensen, K., Jensen, L. N., Rosati, B., Teiwes, R., Daellenbach, K. R., Peräkylä, O., Roldin, P., Bossi, R., Pedersen, H. B., Glasius, M., Bilde, M., and Ehn, M.: Effect of temperature on the formation of highly oxygenated organic molecules (HOMs) from alpha-pinene ozonolysis, Atmos. Chem. Phys., 19, 7609–7625, https://doi.org/10.5194/ACP-19-7609-2019, 2019.

Xavier, C., Rusanen, A., Zhou, P., Dean, C., Pichelstorfer, L., Roldin, P., and Boy, M.: Aerosol mass yields of selected biogenic volatile organic compounds - A theoretical study with nearly explicit gas-phase chemistry, Atmos. Chem. Phys., 19, 13741–13758, https://doi.org/10.5194/ACP-19-13741-2019, 2019.

Ye, Q., Wang, M., Hofbauer, V., Stolzenburg, D., Chen, D., Schervish, M., Vogel, A., Mauldin, R.L., Baalbaki, R., Brilke, S., and Dada, L.: Molecular Composition and Volatility of Nucleated Particles from α-Pinene Oxidation between -50 °C and +25 °C, Environ. Sci. Technol., 53, 12357–12365, https://doi.org/10.1021/ACS.EST.9B03265, 2019.

Zhang, X., Lambe, A. T., Upshur, M. A., Brooks, W. A., Bé, A. G., Thomson, R. J., Geiger, F. M., Surratt, J. D., Zhang, Z., Gold, A., Graf, S., Cubison, M. J., Groessl, M., Jayne, J. T., Worsnop, D. R., and Canagaratna, M. R.: Highly Oxygenated Multifunctional Compounds in α-Pinene Secondary Organic Aerosol, Environ. Sci. Technol., 51, 5932–5940, https://doi.org/10.1021/ACS.EST.6B06588, 2017.

Line 112: unnecessary space in citation

Corrected, thank you.

Table 1 footnote: "based on" instead of "base on"

Corrected, thank you.

Line 175: it is not clear why the experimental conditions cannot be used as initial conditions for GECKO-A?

The GECKO-A simulations were run prior to the chamber experiments. Given the time it takes to run and interpret the GECKO-A simulations, the relative overlap between the GECKO-A simulations and the experimental conditions,
and the co-development of the UCR chamber and the SAPRC box model, we opted for running a greater number of SAPRC simulations and devoting more time to analyzing the SAPRC results.

We have edited the statement **at line 205-208** to more clearly state: "**The GECKO-A simulations were performed for a predefined set of conditions, prior to the chamber experiments, and thus in some cases differ from the experimental conditions.**"

Figure 1: it is difficult to compare the experimental camphene time trends with SAPRC model when they are in separate panels. I suggest overlaying them directly for easier comparison. Same goes for Figure S1.

Thank you for the suggestions. Figure 1 (now Fig. 2 in TM) was updated in the TM (shown below). However, it could be too crowded to overlay three data sets (experimental data with both SAPRC and GECKO-A modeling results) together and make it hard to interpret. Therefore, we decide to keep Fig. S1 in its current form. To achieve an easier
comparison, we modified the y-axis of the figures to get the same scale between different data sets.

[Figure]

**Figure 2. SAPRC predicted β values: (a) without added NOx, and (b) with added NOx. Measured (circles) and predicted (lines) camphene consumption as a function of irradiation time: (c) without added NOx, and (d) with added NOx. The hollow makers used in (c) and (d) are equivalent to dashed lines defined in the legends.**

Line 200-202 and Figure S1. It seems that simulated O3 matches experimental levels in WO experiments, but the trend with increasing HC is inconsistent. SAPRC predicts lower O3 as HC increases, but the experimental trend is more complex. The difference in measured O3 seems quite big between 7ppb and 9ppb experiments, even though the experimental conditions are similar. Predicting O3 in chamber experiments without added NOx is notoriously difficult (e.g. unknown wall outgassing of NOx), so I might be being nitpicky here, but I suggest toning down the sentence

"For all parameters (camphene consumption, NOx decay, O3 formation, and OH levels), the SAPRC simulation results were generally in good agreement with the experimental data."

We agree and thank you for pointing this out. A new sentence was added to **lines 237-240 in TM: "For all parameters (camphene consumption, NOx decay, O3 formation, and OH levels), the SAPRC simulation results were generally in good agreement with the experimental data. The exception to the generally good agreement is O3**

 **predictions in experiments without added NOₓ, which has a relatively strong dependence on the HONO off-gassing rate."**

Figure 7 and Figure S4: After OH addition, the diagram shows that the alkyl radical with a resonance structure (the lone electron is spread over 3 carbons), but I don't think that is true. It is just a tertiary radical.

Thank you for pointing out this mistake. This has been corrected in Fig. 8 in TM and Fig. S4 in TS.

Table 4. VBS parameters: the c* are presumably the c*, not the log of c* (which would be -1,0,1…) If that is the case, the 2nd row should be c* = 1 ug/m3 (not 0)

Corrected, thank you.

Section 4.2 This is a really well written section that shows the most interesting results. It is also nice to see that the change in c* can also be reflected in the VBS parameters. This might be coincidental, but one can see a single alpha
of no added NOx at c* of 10 ug/m3, suggesting dominance of semivolatile material. With NOx, there is a significant amount of nonvolatile material (c* = 0.1ug/m3), and these trends are consistent with the predicted vapor pressures from GECKO-A.

Thank you for the kind words. One point of clarification, the Nannoolal method was used to predict the vapor pressures of the products listed in Table 5, and also is used to calculate the vapor pressures of the products predicted by GECKO-
A, but the products in Table 5 were predicted by SAPRC. Nonetheless, we agree. Though we didn't predict SOA formation using SAPRC, the volatilities of the predicted products under w/ and w/o added NOₓ are reflected in the 2p and VBS fits.

Table 5. What is the definition of "first generation"? Some of these species go through multiple radical intermediates.

To improve clarity, this has been changed to: "1ˢᵗ generation of stable end products formed from camphene reactions
with OH".

Lines 395-399: I am not sure if the argument is clear here. Why does the overall vapor pressure increase with HC0? It is not just partitioning (partitioning does not change the product distribution). Is it linked with RO2 chemistry? i.e. If HC0 increases, then RO2+RO2 increases and RO2+NO decreases, thus less HOMs?

Yes. It is likely linked to the RO₂ chemistry. Starting **line 457**, we've reworded the statement to improve clarity:
**"These trends indicate there is a significant fraction of higher volatility compounds formed that contribute to SOA at higher [HC]₀ (or M₀), resulting in lower SOA mass yields."**

Figure 10: It is interesting that GECKO-A predicts O/C as high as 1.3 at very low HC/NOx, but the AMS did not measure O/C that high. If the authors have time, it would be really nice to see what O/C would look like at HC0/NOx below 1. I do not believe I have ever seen O/C of chamber SOA measured to be 1. But not really a requirement here.
Just curious.

Unfortunately, we do not have any such data for the current set of experiments. That said, we will keep this in mind and measure it in the future if the opportunity arises.

Line 445: it will be really difficult to control beta values in experiments. Previous studies just use a very high NO, but that will shut off the RO2 isomerization channel.

That's true. However, recently experiments performed in our chamber has achieved constant β values from 0 to 1, while maintaining reasonably low NO concentration through the course of the experiments. We found this is out of the scope of the current paper and it was deleted.

**References**

Bianchi, F., Kurtén, T., Riva, M., Mohr, C., Rissanen, M.P., Roldin, P., Berndt, T., Crounse, J.D., Wennberg, P.O.,
Mentel, T.F. and Wildt, J., 2019. Highly oxygenated organic molecules (HOM) from gas-phase autoxidation involving peroxy radicals: A key contributor to atmospheric aerosol. Chemical reviews, 119(6), pp.3472-3509.

Kurtén, T., Rissanen, M.P., Mackeprang, K., Thornton, J.A., Hyttinen, N., Jørgensen, S., Ehn, M. and Kjaergaard, H.G., 2015. Computational study of hydrogen shifts and ring-opening mechanisms in α-pinene ozonolysis products. The Journal of Physical Chemistry A, 119(46), pp.11366-11375.

Molteni, U., Bianchi, F., Klein, F., Haddad, I.E., Frege, C., Rossi, M.J., Dommen, J. and Baltensperger, U., 2018. Formation of highly oxygenated organic molecules from aromatic compounds. Atmospheric Chemistry and Physics, 18(3), pp.1909-1921.

---

## Author Response (AR2)

The authors' edits and responses to both reviewers have substantially improved this manuscript. However, three concerns from the first round of reviews remain only partially addressed, and the manuscript would benefit from more careful consideration of each, as detailed below.

5     We thank the reviewer for further comments and have addressed each additional comment below. Our responses are in blue text, with changes to the manuscript noted in **bold**.

**A. Wall loss:**

**A1.** The additional discussion of wall loss effects remains unconvincing for a number of reasons. First, because this

10     manuscript argues that the chemistry of camphene (especially NOx dependence) is very different from the other precursors for which wall loss has been tested in the Riverside chamber, it can't be expected that wall losses will act similarly. The extent to which vapor wall loss influences measured yields depends on many factors including the distribution of volatilities of SOA-contributing products under all different oxidation conditions -- which may or may not be comparable between camphene oxidation and that of other precursors -- and the speed at which semivolatile

15     vapors are formed, which is likely not comparable to past experiments on systems that used higher [HC]0.

We would first like to emphasize that this is the first time that an explicit gas-phase chemical mechanism has been used to better understand the fate of $RO_2$ radicals in the context of observed SOA formation in the UCR chamber. Therefore, while we can say that the $NO_x$ dependence of SOA formation was the inverse of what is typically observed, we can't say that similar chemistry did not occur in other previous studies in our chamber or in other chambers.

20     We acknowledge that vapor wall losses are highly dependent on the volatility distribution of the oxidation products, which may differ between parent VOCs. We have performed a large number of seeded and unseeded experiments with different parent VOCs in the UCR chamber, yielding oxidation products over a large range of volatilities and at varying generation rates, without reproducing vapor wall loss dependence that has been observed in other studies and described in published literature. There is no evidence that the volatility distribution or the timing of the oxidation

25     products in the camphene experiments were significantly different than the ranges of products previously studied and thus, that more significant vapor wall losses occurred for camphene oxidation. This is further discussed in response to comment #A3 below.

We acknowledge that the systems described above do not include wall loss characterization at low [HC] vapor concentrations and thus the possibility of an underestimation of SOA yield for the low [HC] experiments exists. This

30     is further discussed in response to comment #A3 below.

**A2.** Second, a lack difference between seeded and unseeded experiments does not alone imply the absence of wall effects - see Krechmer et al. 2020 (DOI: 10.1021/acs.est.0c03381).

Regarding the utility of seeded vs. non-seeded experiments for demonstrating wall loss, we thank the reviewer for sharing the Krechmer paper and have cited the reference in the revised manuscript. The theoretical framework and

35    model simulations presented by Krechmer et al. in fact demonstrate the competition between seed and the chamber walls for condensable organics, though it was concluded that high seed concentrations do not negate wall loss effects and an underestimation of SOA yield by 25-35% occurs even when the condensation rate to particles is similar to the condensation rate to chamber walls. These simulation results presented by Krechmer et al. likely overestimate the effects of wall losses, relative to our measured yields for camphene oxidation in the UCR chamber, because: 1) the

40    results are presented for continuous flow experiments which have higher wall loss relative to batch flow experiments, and 2) the wall loss rates are based on those reported for chambers that are 6-20 $m^3$. The UCR chamber used in the camphene oxidation studies was 90 $m^3$, 4.5 times larger than the largest chamber studied in the shared paper, and as noted by Krechmer et al. and others, the wall loss rates are dependent on surface-volume ratio. While additional wall loss characterization for these experiments is not possible, as the UCR chamber has since been rebuilt and reconfigured,

45    we agree with the concluding suggestions by Krechmer et al. that wall loss experiments should be performed for systems characterization, and we will continue to probe this question in the UCR chambers.

**A3.** Third, it's not clear why else these experiments would exhibit such a strong dependence on dHC (and on top of that, why there'd be so much delay in SOA formation which is also dependent on $HC_0$) if not for the influence of wall losses.

50    The dependence of SOA mass yield on $\Delta[HC]$ is expected at low $\Delta[HC]$ due to the non-linear dependence of partitioning on $M_o$; this is assuming consistent chemistry across $\Delta[HC]$ (i.e., an increase in products but not a change in the distribution of products). At a given photochemical aging time (equivalent to the OH level), the amount of $\Delta[HC]$ will increase with $[HC]_0$ and thus the SOA mass yield will also increase. This contributes to an apparent, but not actual, delay in SOA mass/SOA mass yield as a function of $[HC]_0$ shown in Fig. 4. By plotting $M_o$ vs $\Delta[HC]$ in

55    all experiments (Figure S9-with added $NO_x$ and S10-w/o added $NO_x$), it is clear that in the lowest $[HC]_0$ experiments, SOA is formed at very low $\Delta[HC]$, which would not be the case if the majority of the condensable compounds were being lost to the walls. We do not maintain that there is absolutely no wall loss, but that there is no real delay in SOA formation as a function of $[HC]_0$. Based on the shape of the curves (i.e., as described by Ng et al. 2006) there is evidence for oxidation products contributing to SOA mass that are being formed more slowly than camphene is being

60    consumed, which is consistent with the mechanistic description presented in the manuscript (see Fig. S5).

[Figure]

**Figure S9. Measured SOA mass concentrations as a function of reacted camphene concentration with added NO$_x$; inset shows the lowest camphene concentrations from 0 – 400 μg m$^{-3}$.**

[Figure]

**Figure S10. Measured SOA mass concentrations as a function of reacted camphene concentration without added NO$_x$; inset shows the lowest camphene concentrations from 0 – 400 μg m$^{-3}$.**

We have added the following text referencing the new SI figures (line 288):

"In the camphene experiments presented herein, the aging effects were determined to be less important than RO$_2$ chemistry, since the SOA mass yield curves as a function of photochemical aging already plateau or nearly plateau by the end of experiments (Fig. 4) and **the shapes of the growth curves (Fig. S9 and Fig. S10) indicate different kinetics and contributions from oxidation products that form slowly among and between the experiments with and without added NO$_x$ (Ng et al., 2006)."**

**A4.** All these reasons merit stronger caveats on the results presented herein. Most importantly, although it is suggested that vapor wall loss "would not affect the following discussion or conclusions regarding the role of RO2 chemistry", this is not entirely accurate; if wall-loss correction of SOA yields in experiments W1-2, for which the correction will

be largest out of the high-NOx experiments, were sufficiently large, they might counteract the observed decrease in yield at "extreme NOx", or at least render them statistically insignificant.

In response to comments A1-A4, and in an effort to make our assumptions clearer, we have made the following changes to the manuscript.

We have revised the discussion of vapor wall loss under section 2.1 (Chamber Facility and Instrumentation) to read as follows:

**Particle wall loss corrections were performed following the method described in Cocker et al. (2001). Vapor wall loss of organics has been reported in multiple chambers (e.g., Zhang et al., 2015, 2014; Schwantes et al., 2019); and has been modeled as a function of the mass and volatility of the condensing compounds, condensation sink, and characteristics of the chamber (e.g., La et al., 2016; Zhang et al., 2014; Ye et al., 2016) . The extent to which these observations and modeling simulations are relevant in the UCR chamber is unclear, given the significant difference in chamber sizes. The UCR chamber is 4.5 times larger (90 m$^3$) than the largest referenced chamber in these studies (20 m$^3$) and most are ~10 m$^3$. In the UCR chamber, the role of vapor wall loss has been investigated in SOA experiments using various precursor compounds (including α-pinene and *m*-xylene) under seed and no seed conditions (Clark et al., 2016; Li et al., 2015). There has been no evidence of non-negligible vapor wall loss in those experiments, including no measurable differences in SOA formation in experiments with and without seed. In this work, stability tests on camphene demonstrated negligible vapor wall loss of the parent compound. Thus without evidence to suggest otherwise, negligible vapor wall loss was assumed for these experiments. This assumption is further discussed where it may affect the major conclusions regarding the role of gas-phase chemistry on SOA formation.**

We have also added the following caveats:

- In the presentation of the SOA mass and yield (section 3.2), line 251: **The SOA mass yields measured at the lowest [HC]$_0$/ Δ[HC] may be an underestimate due to the assumption of negligible vapor wall loss, which would have the largest effect at low [HC] (Krechmer et al., 2020).**

- In the discussion of Fig. 5: "Only over low Δ[HC] ranges, the SOA mass yield increases with Δ[HC] in experiments without added NO$_x$ due to the dependence of partitioning on $M_o$ (or Δ[HC]). **This trend may be exaggerated due to the assumption of negligible vapor wall loss, which could result in an underestimation of SOA mass yield particularly at low Δ[HC] (Krechmer et al., 2020)."**

We have additionally revised Fig. 5, which we think more clearly shows the following: 1) the role of Δ[HC] at low Δ[HC] particularly for the experiments without added NO$_x$; 2) the generally higher SOA yields in experiments with added NO$_x$; and 3) the generally higher SOA yields at "moderate" [HC]$_0$/[NO$_x$]$_0$ values.

[Figure]

**Figure 5. SOA mass yield (color bar) as a function of Δ[HC], $[HC]_0/[NO_x]_0$, and photochemical aging time, with added $NO_x$ experiments square markers and without added $NO_x$ experiments round markers.**

The text associated with Fig. 5 has been modified to read: "Over low Δ[HC], SOA mass increased in experiments without added $NO_x$ due to the dependence of partitioning on $M_o$ (or Δ[HC]). This trend may be exaggerated due to the assumption of negligible vapor wall loss, which could result in an underestimation of SOA mass yield particularly at low Δ[HC] (Krechmer et al., 2020). The sensitivity of SOA formation to $[HC]_0/[NO_x]_0$ over the range of [HC] sampled is also shown. At a given Δ[HC] level, a lower $[HC]_0/[NO_x]_0$ (when within 0.5–200) led to a higher SOA mass yield (decreasing $[HC]_0/[NO_x]_0$ by ~2 orders of magnitude resulted in a factor of two increase in SOA mass yield). **The chamber data presented here indicate that the highest SOA mass yields from camphene were observed in a regime of high Δ[HC] and moderate $[HC]_0/[NO_x]_0$; this regime is distinguished from an extreme $[NO_x]$ regime, proposed in section 4.2, in which SOA mass yields are suppressed at the lowest $[HC]_0/[NO_x]_0$ (also shown in Fig. 5).** These observations are different from those in studies of α-pinene, in which lower $[HC]_0/[NO_x]_0$ generally led to lower SOA mass yield (Eddingsaas et al., 2012). **The observed trends are further explored in the following sections, particularly the role of $RO_2$ based on SAPRC simulations.**"

Regarding experiments W1 and W2 and the existence of an extreme $NO_x$ regime, we provide the following analysis. In the table below, the with added $NO_x$ experiments are ordered from low to high $[HC]_0/[NO_x]_0$. It can be seen that for experiments W3-W7, SOA mass yield decreases as $[HC]_0/[NO_x]_0$ increases. As has been described in the manuscript, this trend is plausibly explained by the SAPRC modeling simulations. Experiments W1 and W2 are clearly outliers and do not follow the same trend. If we assume, as arguably an extreme upper limit, the observed SOA mass concentrations in W1 and W2 were ~25% low (using the estimate in Krechmer et al.), the calculated SOA mass yields increase to 46% and 41%, respectively, which is not enough to negate our conclusions regarding an extreme $NO_x$ regime.

| Expt. | Δ[HC] (µg m$^{-3}$) | $[HC]_0/[NO_x]_0$ (ppb/ppb) | SOA Mass Yield |
|---|---|---|---|
| W1 | 40 | 0.08 | 0.36 |
| W2 | 140 | 0.18 | 0.33 |
| W3 | 177 | 0.51 | 0.64 |
| W5 | 334 | 0.64 | 0.60 |
| W6 | 724 | 1.33 | 0.59 |
| W7 | 950 | 2.88 | 0.52 |
| W4 | 237 | 5.91 | 0.41 |

We have added the following in 4.2, before the description of the extreme NO$_x$ regime: "The relatively low SOA mass yields in experiments W1 and W2 (0.36 and 0.33) also can be explained by differences in product distribution. **An underestimation of the SOA mass yields in these experiments due to the assumption of negligible wall loss is not sufficient to explain these relatively low yields**."

**B. Uncertainty:** the +/- 6.65% experimental uncertainty on an individual experiment's measured yield is a useful number and good to have included in the revisions. What's missing now is any discussion of how that uncertainty might affect the conclusions drawn here. Given that uncertainty range, for example, it does not necessarily seem that the decreased SOA yield observed at high SOA mass under both high- and low-NOx conditions (the purple points in Figure 2) can be statistically significantly distinguished from the plateaus (the yellow, light blue, and dark blue points). It seems from my reading (and without doing any quantitative statistical analysis) that this is the only place where the uncertainty bounds may be large enough to affect the significance of the conclusions, but it would be reassuring if the authors could comment further on statistical significance throughout.

Uncertainty in observed SOA mass and calculated SOA mass yields is often approximated by the cumulative uncertainty in the measured quantities and sometimes by performing replicate experiments (e.g., during characterization). The uncertainty reported here is of the latter type, but was calculated for other experimental systems; thus using that reported value for quantitative statistical analysis would not necessarily result in a robust result. We agree with the reviewer that the decreased yields at high ΔHC/ SOA mass may not be statistically different than the plateaus but because we saw the same behavior in the GECKO-A simulations, we thought it was worth exploring with the SAPRC model. The SAPRC modeling results provide a plausible explanation, though as with any model simulations, further experiments should be done to confirm and constrain the predictions. We have made an effort to make the above points clearer through the following modifications.

- We have revised the text in section 2.1 to read: "**A prior characterization of this UCR chamber system (Li et al., 2016) reported an experimental uncertainty in SOA yields of < 6.65%.**"

- In section 3.2, after the Fig. 3 description we have added: **"The difference between the peak SOA mass yield and the SOA mass yield at the highest $[HC]_0$ was ~0.12 with added $NO_x$ and ~0.08 without added $NO_x$. While the SOA mass yields at the highest $[HC]_0$ may not be statistically different within the uncertainty of the measurements, this trend was also observed in the GECKO-A model simulations (see Sect. 5) and thus was further investigated, and reasonably explained, by the $RO_2$ fate based on box model simulations (see Sect. 4 & 5)."**

- In section 4.2, line 413: **"While the decreasing SOA mass yields at high $[HC]_0$ and $M_0$ in experiments with and without added $NO_x$ (shown in Fig. 3) may not be statistically different within the uncertainty of the measurements, $RO_2$ chemistry was explored as an explanation for the apparent trends."** In all occurrences, "led to" was replaced by "can explain" to better convey that while the observations can be explained by the SAPRC model predictions further experiments are needed to confirm and constrain.

**C. Figure 3 (now 4):** Putting these on the same axes does indeed make it easier to see that at any given point, the high-NOx yield is higher than the low-NOx yield, and it's good this point was highlighted in the text. However, it does not convincingly show that the yields have all plateaued or nearly plateaued by the end of the experiment, or that the trends would remain as pronounced as you show if the low-NOx experiments were allowed to run longer. In fact, the new figure draws greater attention to the delay in SOA formation in low-HC experiments -- and whether or not this is a consequence of wall losses (see above), it merits some explanation. It still seems likely that the yields is low-NOx experiments would be higher than reported if they were allowed to run longer. A similar comparison across experiments to Figure 3 (now 4) but in the growth curve space (Delta[M(0)] vs Delta[HC], e.g. Ng et al., ES&T, 2006, DOI 10.1021/es052269u) might help to distinguish whether the different shapes of these curves imply differences in chemistry or just in oxidation extent, and help to visualize the remaining SOA that might be formed if the experiments were allowed to continue.

We thank the reviewer for the suggestion to plot $M_0$ vs. $\Delta[HC]$. Two new figures were created and added to the SI (Fig. S9 and Fig. S10). The figures are shown and discussed in the response to comment A3.

Regarding the SOA mass yield curves as a function of photochemical aging time, we have revised the manuscript to read:

"Experiments with added $NO_x$ generally had longer photochemical aging times than experiments without added $NO_x$; without added $NO_x$, **all experiments may not have fully plateaued and thus would have had higher SOA mass yields at longer irradiation times**. However, even at the same aging time (Fig. S8), the SOA yields were higher in the experiments with added $NO_x$."

Fig. 5 illustrates that SOA yield is not only dependent on $\Delta[HC]$, but also on $[HC]_0/[NO_x]_0$. Beyond the differences in OH levels, as discussed in the manuscript, this is indicative of the $RO_2$ chemistry changing with $\Delta[HC]$, particularly the fates of the $RO_2$ radicals as a function of $[HC]_0/[NO_x]_0$. The different shapes of the curves in Fig. S9 and Fig. S10

190     are consistent with our conclusions that the chemistry is different among the experiments with added $NO_x$ and without added $NO_x$, and between the with and without added $NO_x$ experiments.

---

## Author Response (AR3)

**Editor**: Thank you for your response to the further comments made by Reviewer 1. I just have one quick clarifying question before accepting the manuscript for publication. In the response to comment A3, it is noted that "there is no real delay in SOA formation as a function of [HC]0." However, Figures S9 and S10 show that there is a delay in SOA formation as a function of [HC]0, in which experiments with higher [HC]0 have a larger delta[HC] before any SOA formation is observed?

Dear Dr. Ng,

You are correct. The reviewer was concerned about wall loss affecting SOA yields at low $[HC]_0$ or $\Delta[HC]$ and suggested these plots as a way of visualizing a potential effect. As you note, if anything, there is a delay at higher $[HC]_0$ in that it takes a larger delta HC to start forming SOA. This is not related to wall loss, but OH levels in the higher $[HC]_0$ experiments. We responded to the reviewer in the context of the wall loss question. It is more accurate to say that "there is no real delay in SOA formation at low $[HC]_0$ that would be suggestive of vapor wall loss".

Thank you for your attention to detail in handling this manuscript. Since no edits were made to the manuscript, the last submitted version 3 was submitted again with this reply.